# Mapping vascular network architecture in primate brain using ferumoxytol-weighted laminar MRI

**Joonas A Autio**[1,2,3]*, **Ikko Kimura**[1], **Takayuki Ose**[1], **Yuki Matsumoto**[1], **Masahiro Ohno**[1], **Yuta Urushibata**[4], **Takuro Ikeda**[1], **Matthew F Glasser**[2,3], **David C van Essen**[3], **Takuya Hayashi**[1]

[1]Laboratory for Brain Connectomics Imaging, RIKEN Center for Biosystems Dynamics Research, Kobe, Japan; [2]Department of Radiology, Washington University in St. Louis, St. Louis, United States; [3]Department of Neuroscience, Washington University in St. Louis, St. Louis, United States; [4]Siemens Healthcare K.K., Tokyo, Japan

## eLife Assessment

This study presents **valuable** findings on the relative cerebral blood volume of non-human primates that move us closer to uncovering the functional and architectonic principles that govern the interplay between neuronal and vascular networks. The evidence of areal variations and of vessel counting and laminar analysis is **solid**. The lack of a direct comparison of their approach against better-established MRI-based methods for measuring hemodynamics and vascular structure somewhat weakens the evidence provided in the current paper version, but the current work is an significant step forward. The work will be of interest to NHP imaging scientists.

*For correspondence:
autio@wustl.edu

**Abstract** Mapping the vascular organization of the brain is of great importance across various domains of basic neuroimaging research, diagnostic radiology, and neurology. However, the intricate task of precisely mapping vasculature across brain regions and cortical layers presents formidable challenges, resulting in a limited understanding of neurometabolic factors influencing the brain's microvasculature. Addressing this gap, our study investigates whole-brain vascular volume using ferumoxytol-weighted laminar-resolution multi-echo gradient-echo imaging in macaque monkeys. We validate the results with published data for vascular densities and compare them with cytoarchitecture, neuron and synaptic densities. The ferumoxytol-induced change in transverse relaxation rate ($\Delta R_2^*$), an indirect proxy measure of cerebral blood volume (CBV), was mapped onto 12 equivolumetric laminar cortical surfaces. Our findings reveal that CBV varies threefold across the brain, with the highest vascular volume observed in the inferior colliculus and lowest in the corpus callosum. In the cerebral cortex, CBV is notably high in early primary sensory areas and low in association areas responsible for higher cognitive functions. Classification of CBV into distinct groups unveils extensive replication of translaminar vascular network motifs, suggesting distinct computational energy supply requirements in areas with varying cytoarchitecture types. Regionally, baseline $R_2^*$ and CBV exhibit positive correlations with neuron density and negative correlations with receptor densities. Adjusting image resolution based on the critical sampling frequency of penetrating cortical vessels allows us to delineate approximately 30% of the arterial–venous vessels. Collectively, these results mark significant methodological and conceptual advancements, contributing to the refinement of cerebrovascular MRI. Furthermore, our study establishes a linkage between neurometabolic factors and the vascular network architecture in the primate brain.

## Introduction

The brain's vascular network plays a crucial role in delivering oxygen, glucose, and other nutrients while clearing metabolic by-products to meet the high energy demands of neural information processing. Understanding the organization of the brain's vasculature is vital for diagnosing and addressing clinical deficits related to stroke, vascular dementia, and neurological disorders with vascular components (*Iadecola, 2013*; *Sweeney et al., 2018*; *Toledo et al., 2013*). Furthermore, it is essential for advancing the applications of functional MRI (fMRI), as vascular density has implications for statistical power, and the arrangement of large vessels may impose limitations and biases on the spatial accuracy of functional localization. Despite its significance, our knowledge of the vascular network architecture in the primate cerebral cortex remains limited (*Duvernoy et al., 1981*; *Schmid et al., 2019*; *Weber et al., 2008*).

Anatomically, blood flows from the pial vessel network via feeding arteries and arterioles to capillary beds in each cortical layer, ultimately leading via draining veins back to the pial vessel network. The capillary density varies with the rate of oxidative metabolism across cortical layers and exhibits sharp transitions between some cortical areas (*Duvernoy et al., 1981*; *Ji et al., 2021*; *Zheng et al., 1991*). Recent advances in immunolabeling and tissue clearing techniques have enhanced our understanding of brain vascularity in post-mortem mouse brains (*Ji et al., 2021*; *Kirst et al., 2020*). These studies have demonstrated heterogeneous vasculature varying in capillary length density threefold across brain regions and moderate variation across cortical layers. Still, methodological and analytical challenges have limited quantitative anatomical research in primates to a small number of cortical regions (*Harrison et al., 2002*; *Lauwers et al., 2008*; *Weber et al., 2008*) and quantitative anatomical research requires investigation across a broader range of cortical regions in primates.

Mapping the brain-wide vasculature using MRI faces several challenges due to the intricate nature of the vascular network. One crucial criterion for successful vascular mapping is arterio-venous density, which is necessary to delineate individual large-caliber vessels from microcapillary networks. The combined surface density of intra-cortical feeding arteries and draining veins is about 7 vessels/$mm^2$ (*Weber et al., 2008*). According to the sampling theorem, this implies that the minimal (spatial) sampling frequency is $\approx$14 voxels/$mm^2$ ($\approx$0.26 mm isotropic) imposing stringent image acquisition requirements to critically sample cortical vasculature. Ferumoxytol contrast agent-weighted MRI offers a safe and indirect means to measure relative vascular volume and enhance the visibility of large vessels (*Boxerman et al., 1995*; *Kim et al., 2013*; *Muehe et al., 2016*; *Yablonskiy and Haacke, 1994*). Compared to clinically used gadolinium-based agents, ferumoxytol's substantially longer half-life and stronger $R_2$* effect allows for higher resolution and more sensitive vascular volume measurements (*Buch et al., 2022*), albeit these methodologies are hampered by confounding factors such as vessel orientation relative to the magnetic field ($B_0$) direction (*Ogawa et al., 1993*).

The macaque monkey is an excellent experimental non-human primate model to objectively investigate the MRI resolution and contrast requirements and their limitations for mapping arterio-capillary-venous networks. Quantitative vascular density data is available for a limited number of cortical areas (*Weber et al., 2008*), providing essential insights for determining vessel-density informed minimum image resolution requirements. Importantly, experiments in macaque monkeys can also help elucidate the neurometabolic factors that shape vascular network architecture. For instance, variations in the cellular composition (*Collins et al., 2010*), synaptic density (*Elston, 2002*), receptor distribution (*Froudist-Walsh et al., 2023*), neural connectivity (*Felleman and Van Essen, 1991*; *Markov et al., 2014a*), myelination (*Lewis and Van Essen, 2000*), and oxidative metabolism (*Sincich et al., 2003*) are well documented, but the relationships between these factors and vascular architecture have only been investigated in a few cortical areas (*Borowsky and Collins, 1989*; *Tsai et al., 2009*; *Weber et al., 2008*).

In this study, we used ferumoxytol contrast agent-weighted 3D multi-echo gradient-echo MRI to investigate vascular heterogeneity in macaque monkey brains. We scaled image resolution to meet specific requirements, aiming to delineate cortical layers and individual vessels. Using this advanced MRI and laminar surface mapping, we then elucidate neuroanatomical factors underlying the heterogeneous vasculature. Our analysis reveals insights into translaminar and regional heterogeneities, and signatures of neuroanatomical organization within the macaque cerebral cortex. By addressing these dual objectives—advancing vascular MRI technology and uncovering the neuroanatomical factors shaping cortical vascularity—we contribute to both methodological and conceptual advancements in

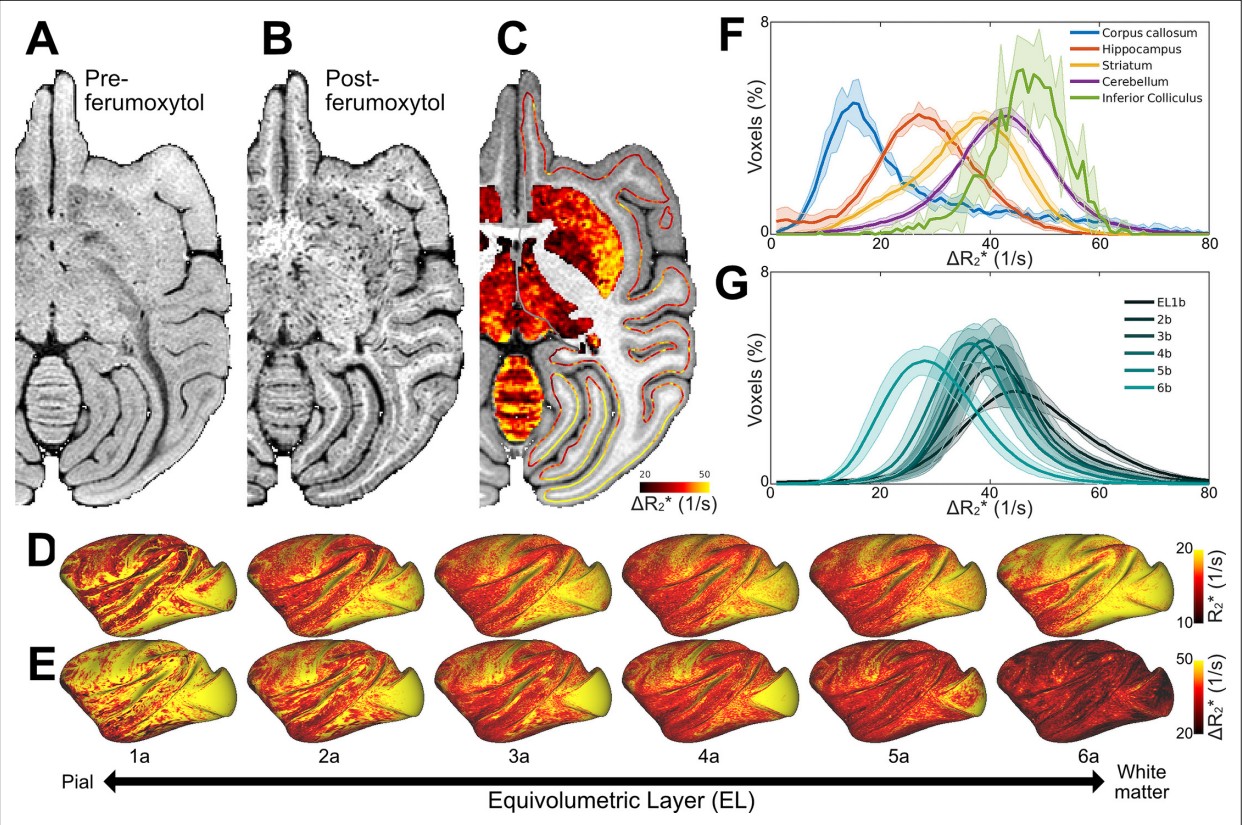

**Figure 1.** Ferumoxytol-weighted MRI reveals heterogeneous vascularity in the macaque brain. Representative 3D gradient-echo images (**A**) before and (**B**) after the ferumoxytol contrast agent injection. (**C**) Ferumoxytol-induced change in transverse relaxation rate ($\Delta R_2^*$) displayed on subcortical gray matter and cortical midthickness surface contour ($N = 1$). Average (**D**) pre-ferumoxytol $R_2^*$ and (**E**) $\Delta R_2^*$ equivolumetric layers (ELs; $N = 4$). (**F**) Histograms in selected brain regions and (**G**) ELs. Solid lines and shadow indicate mean and standard deviation ($N = 4$), respectively.

The online version of this article includes the following figure supplement(s) for figure 1:

**Figure supplement 1.** Transverse relaxation rate ($R_2^*$) measures are biased by the orientation of the static magnetic field ($B_0$).

the field. Our findings offer not only a framework for objectively validating cerebrovascular MRI but also a deeper understanding of how neural and vascular systems intricately interact in the primate cerebral cortices.

## Results

### Laminar $R_2^*$ and ferumoxytol-induced $\Delta R_2^*$ MRI in macaque cerebral cortex

*Figure 1* displays representative gradient-echo images before (*Figure 1A*) and after (*Figure 1B*) intravascular ferumoxytol injection ($N = 1$). The ferumoxytol effectively reversed the signal-intensity contrast

**Table 1.** Estimated transverse relaxation rate ($R_2^*$) before and after injection of ferumoxytol contrast agent.

Values are mean (std) ($N = 4$). Abbreviation: WM: white matter.

| | $R_2^*$ [s⁻¹] | Ferumoxytol $R_2^*$ [s⁻¹] | $\Delta R_2^*$ [s⁻¹] | Vascular volume (%) |
|---|---|---|---|---|
| Cortex | 17.5 ± 0.6 | 56.4 ± 1.3 | 38.9 ± 1.4 | 2.0–2.2* |
| WM | 21.6 ± 0.6 | 45.1 ± 1.0 | 23.5 ± 0.7 | 0.9–1.2* |
| Ratio | 0.8 ± 0.1 | 1.3 ± 0.1 | 1.7 ± 0.1 | 1.8–2.1* |

*__Weber et al., 2008__.

between gray matter and white matter while enhancing the visibility of large vessels, as expected. For quantitative assessment, $R_2*$ values were estimated from multi-echo gradient-echo images acquired both before and after the administration of ferumoxytol contrast agent (**Table 1**). Subsequently, the baseline $R_2*$ and $\Delta R_2*$, an indirect proxy measure of CBV (**Boxerman et al., 1995**), volume maps for each subject were mapped onto the 12 native equivolumetric layers (ELs) (**Figure 1C**). Each vertex was then corrected for normal of the cortex relative to $B_0$ direction (**Figure 1—figure supplement 1A–C**). Surface maps for each subject were registered onto a Mac25Rhesus average surface using cortical curvature landmarks and then averaged across the subjects (**Figure 1D, E**). Around cortical midthickness, the distribution of $R_2*$, an aggregate measure for ferritin-bound iron, myelin content and venous oxygenation levels (**Langkammer et al., 2012**), resembled the spatial pattern of $\Delta R_2*$ vascular volume. However, across cortical layers, these measures exhibited reversed patterns: $R_2*$ increased toward the white matter surface whereas $\Delta R_2*$ decreased (**Figure 1D, G**).

To explore heterogeneous brain vascularity, we investigated $\Delta R_2*$ in selected subcortical regions (**Figure 1F**). We found the highest CBV in the inferior colliculus, an early auditory nucleus, and the lowest in the corpus callosum. Overall, the relative blood volume variations among the investigated subcortical regions were comparable to those reported in mice (**Kirst et al., 2020**).

Adjacent to the pial surface, large vessels exhibited notable signal-loss in the superficial gray matter. To visualize the pial vessel network, we removed very low-frequency components from TE-averaged post-ferumoxytol signal-intensity maps and identified continuous signal dropouts along ELs using clustering. Within the most superficial layers (e.g., EL1a–2a), this analysis revealed an extensive arterio-venous pial vessel network spanning almost the entire cortical surface (**Figure 2A**). We attempted to delineate pial arteries and veins using pre-contrast $R_2*$ values; however, due to the 'blooming' effect of ferumoxytol (**Buch et al., 2022**) distinguishing adjacent large-caliber vessels was difficult to differentiate with high confidence. Additionally, the continuity of the pial vessel network may also have been influenced by veins crossing the sulci (**Duvernoy et al., 1981**). Pial vessel network was consistently observed across subjects (**Figure 2—figure supplement 1A**), although the precise locations of vessels did vary across subjects. In contrast, in the middle and deep cortical layers (EL2b-6b) continuous signal-dropout clusters were largely absent demonstrating that the influence of the large pial vessels was minimal in these layers (**Figure 2—figure supplement 1B**).

To visualize the intra-cortical vessel network, we next performed ferumoxytol-weighted experiments with isotropic image resolution of 0.23 mm adjusted below to the critical (spatial) sampling frequency of large penetrating vessels (19 vs 7 vessels/mm$^2$) (**Zheng et al., 1991**; **Weber et al., 2008**). The post-ferumoxytol signal-intensity maps (**Figure 2—figure supplement 2A**) were used to identify vessels in volume space using the Frangi filter (**Figure 2—figure supplement 2B**) and in the cortical surface by calculating sharp gradients and determining their local minima (**Figure 2—figure supplement 2C**). Local minima, however, by mathematical definition can capture 1 vessel per 7 vertices (each vertex contains six neighbors). To address this limitation, we generated an ultra-high cortical surface mesh (656k) with an average vertex area of 0.022 ± 0.012 mm$^2$ (1st–99th percentile range: 0.006–0.065 mm$^2$) (**Figure 2B**). Within the cortical gray matter, we identified an average of 24,000 ± 2000 penetrating vessels per hemisphere (**Figure 2B, C**; for more ELs see **Figure 2—figure supplement 3**). In V1, we found 1.9–2.2 vessels/mm$^2$ using Frangi filter and surface vessel detection, respectively (**Figure 2E**). This vessel density corresponds to about 30% of the anatomical ground-truth (**Zheng et al., 1991**; **Weber et al., 2008**).

To corroborate the periodicity of the cerebrovascular network, we next applied a non-uniformly sampled Lomb–Scargle geodesic periodogram analysis on the signal-intensity averaged native ELs (**Figure 2D**). The periodograms revealed dominant periodicity at spatial frequency of ≈0.6 1/mm in the most superficial ELs, likely reflecting the presence of large-caliber vessels in the pial network. In the middle ELs, bimodal distribution was observed with peaks at around 0.6 and 1.2 1/mm. In the deep ELs, peak power occurred at a shorter distance (≈1 1/mm), potentially indicative of the large arteries supplying the white matter. These findings underscore the substantial variation in vascular organization across the cortical layers.

To explore areal differences in translaminar features, we next parcellated the dense $R_2*$ and $\Delta R_2*$ maps using M132 cortical atlas (**Figure 2A–D**). To mitigate bias resulting from undersampling the large-caliber vessels, median parcel values were used for parcellation, $\Delta R_2*$ profiles were detrended across ELs and then averaged across subjects. In the EL4b, which approximately corresponds to the

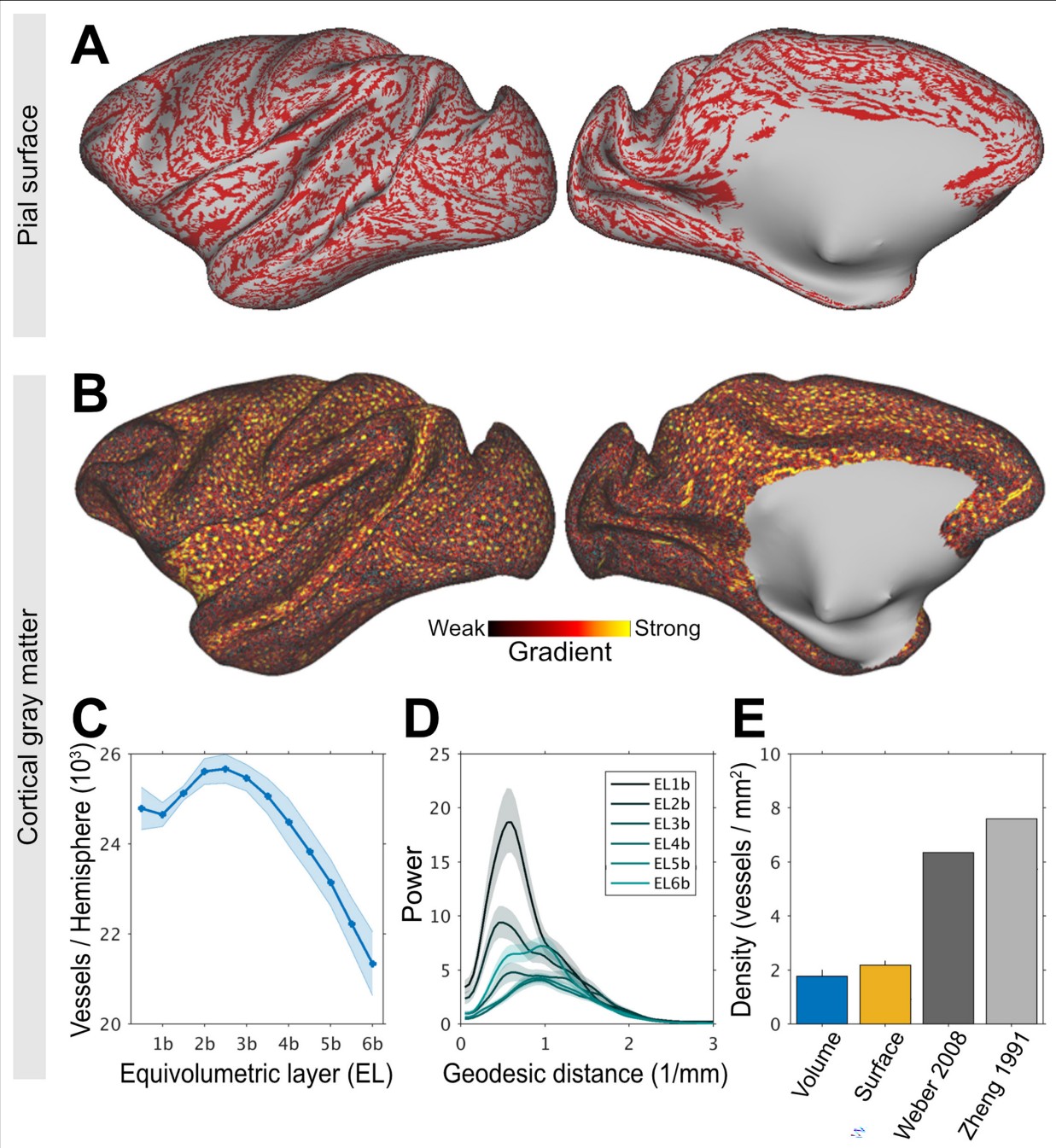

**Figure 2.** Charting large-caliber vessel networks in the cerebral cortex. (**A**) Ferumoxytol-weighted MRI reveals a continuous pial vessel network running along the cortical surface. Note that the large vessels branch into smaller pial vessels. (**B**) Cortical surface mapping of intra-cortical vessels. Vessels were identified using high-frequency gradients (red-yellow colors) and each blue dot indicates the vessel's central location. Representative equivolumetric layer (EL) 4a is displayed on a 656k surface mesh. (**C**) Number of penetrating vessels across ELs per hemisphere. Solid lines and shadow show mean and standard deviation across TEs ($N = 1$). (**D**) Non-uniformly sampled Lomb–Scargle geodesic-distance periodogram. The vessels exhibit a peak frequency at about 0.6 1/mm reflecting the frequency of large-caliber vessels. (**E**) Comparison of vessel density in V1 determined using MRI (current study) and 'ground-truth' anatomy. In volume space, the density of vessels was estimated using Frangi filter whereas in the surface mesh the density was estimated using local minima. These are compared to the density of penetrating vessels with a diameter of 20–50 µm (**Zheng et al., 1991**) and the density of feeding arterial and draining veins evaluated using fluorescence microscopy (**Weber et al., 2008**).

The online version of this article includes the following figure supplement(s) for figure 2:

**Figure supplement 1.** Consistency of pial vessel network mapping across subjects.

**Figure supplement 2.** Charting vessels in the visual cortex.

**Figure supplement 3.** Detection of intra-cortical vessels across equivolumetric layers.

location of histologically defined thalamic input L4c, the primary visual area exhibited larger vascular volume in comparison to surrounding cortical areas (*Figure 3B, D*). Within the visual system, inspection of laminar profiles revealed distinct features.

Since the $\Delta R_2^*$ is an indirect proxy measure of vascular volume (*Boxerman et al., 1995*), we next sought to validate the noninvasive laminar $\Delta R_2^*$ maps with respect to quantitative histological assessment of vascular properties in the macaque visual cortex (*Weber et al., 2008*). In V1 we found that $\Delta R_2^*$ more closely resembled microcapillary and oxidative metabolism rather than large vessel volume fraction (*Figure 3F*), albeit we could not identify the vascularity peak in L6 potentially due resolution limitations. Moreover, the V2/V1 $\Delta R_2^*$ ratio in EL4b (79 ± 5%) (*Figure 3E*) was also in excellent agreement with the previous reports of capillary volume (*Zheng et al., 1991*; *Weber et al., 2008*). Finally, the average $\Delta R_2^*$ ratio between V1 gray matter and the underlying white matter (2.2 ± 0.1) was also close to the histological assessment (1.8–2.1). Taken together, we found comparable relative variations in vascular volume with anatomical 'ground-truth' substantiating the validity of our noninvasive methodology.

## Variations in cerebrovascular network architecture reveal inter-areal boundaries

Since cellular composition (*Collins et al., 2010*) and oxidative metabolism (*Sincich et al., 2003*) are known to exhibit sharp transitions between cortical areas, we next tested the hypothesis whether the variations in vascular network architecture may also reveal inter-areal boundaries (*Zheng et al., 1991*). To address this question, we calculated the gradient-ridges of $\Delta R_2^*$ in each EL (*Figure 1E*). Due to the strong cortical contrast ($\Delta R_2^* = 39 ± 2$ ms), the resulting gradients were notably strong and revealed several sharp transitions (Figure 5A, B). A particularly strong gradient was observed at the boundary between V1/V2 at EL4b (*Figures 3D and 4A, B*), attributable to the relatively large capillary density difference between the areas (*Figure 3E*; *Duvernoy et al., 1981*; *Zheng et al., 1991*; *Weber et al., 2008*). We also found a sharp $\Delta R_2^*$ transition between the primary sensory cortex (area 3) and the primary motor cortex (area 4), in line with histological evaluation of capillary density in humans (*Duvernoy et al., 1981*). Moreover, we discovered a sharp $\Delta R_2^*$ transition between area 3 and the secondary sensory area (Brodmann area 2). The estimated area boundary locations were supported by comparison of cortical area boundaries as defined in the M132 atlas (*Figure 4A, B*; *Markov et al., 2014b*). The auditory cortex exhibited also relatively high $\Delta R_2^*$, however, the gradient-ridges were less distinguishable in this region. Multiple layer-specific gradient-ridges were also observed, albeit these were weaker in magnitude and more challenging to delineate.

Because terminals of myelinated axons often overlap with high oxidative metabolism (*Horton, 1984*; *Rockoff et al., 2014*), we also examined the association between $\Delta R_2^*$ and T1w/T2w-FLAIR, an indirect proxy measure of cortical myelin density (*Figure 4A, C*; *Glasser and Van Essen, 2011*; *Autio et al., 2024*). We found that M132 atlas parcellated $\Delta R_2^*$ was positively correlated with intra-cortical T1w/T2w-FLAIR myelin ($R = 0.49 ± 0.16$), and negatively correlated with cortical thickness ($R = –0.37 ± 0.10$) (*Figure 4D*).

## Translaminar vascular volume variations link with neuroanatomical organization

Across the cortical areas and layers, average $\Delta R_2^*$ profiles exhibited moderate variability (*Figure 5A*, *Figure 3—figure supplement 1*). To search for repetitive patterns in translaminar vascularity, we applied agglomerative clustering to concatenated group data. This analysis revealed distinct groups of vascularity arranged between eulaminate and agranular regions (*Figure 5B–D*). A unique vascular profile was identified in V1, characterized by very dense vascularity and prominent peak density in EL4b (*Figure 5A, E*). Average cluster profiles demonstrate that translaminar $\Delta R_2^*$ was relatively high in isocortical areas, and low in agranular areas (*Figure 5E*). The cluster boundaries (*Figure 5D*) typically occurred in vicinity of the strong $\Delta R_2^*$ gradients (*Figure 4B*). Given the clustering between agranular and granular eulaminate cortices, we further corroborated whether the heterogeneous vascularization is associated with local microcircuit specialization of the cortex. For this objective, we used the designation of the cytoarchitectonic classification mapped onto M132 atlas (*Burt et al., 2018*; *Hilgetag et al., 2016*). This analysis confirmed that CBV, indeed, varies along the cytoarchitectonic types (Kendall's tau $\tau = 0.69$, $p < 10^{-5}$) (*Figure 5F*). We also found a close association between baseline $R_2^*$

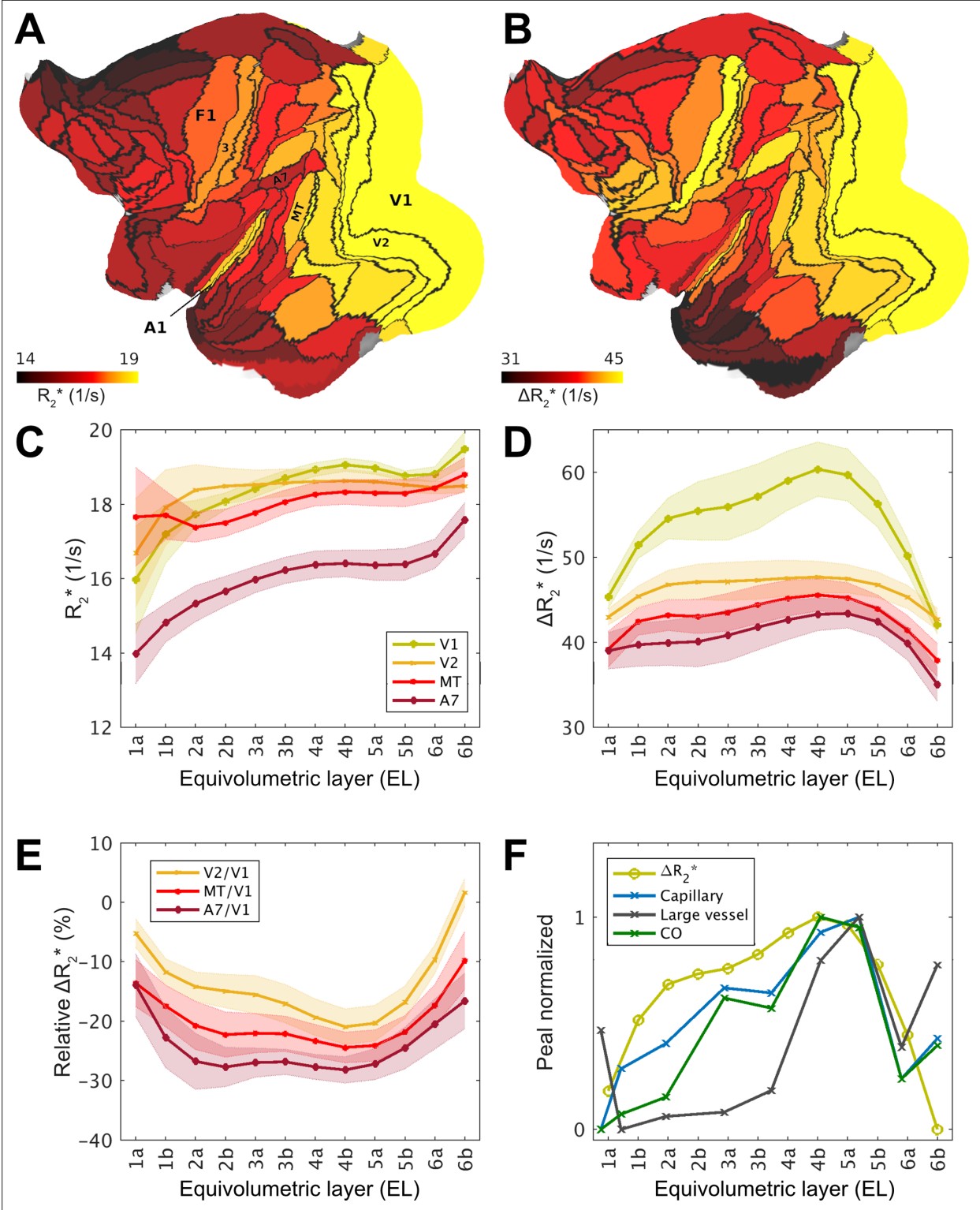

**Figure 3.** Exemplar laminar profiles of transverse relaxation rate ($R_2^*$) and ferumoxytol-induced change in $R_2$ ($\Delta R_2^*$) in the macaque cerebral cortex. (**A**) Exemplar equivolumetric layer 4b (EL4b) $R_2^*$ and (**B**) $\Delta R_2^*$ displayed on cortical flat-map. Note that primary sensory areas (e.g., V1, A1, and area 3) and association areas exhibit high and low $\Delta R_2^*$, respectively. (**C, D**) Exemplar laminar profiles from visual cortical areas. Solid lines and shadow show mean and standard deviation across hemispheres, respectively. (**E**) Laminar $\Delta R_2^*$ profiles relative to the V1. Solid lines and shadow show mean and inter-subject standard deviation. (**F**) Peak-normalized $\Delta R_2^*$ profile compared with anatomical ground-truth in V1 (*Weber et al., 2008*). Cytochrome-c oxidase

*Figure 3 continued on next page*

(CO) activity, capillary and large vessel volume fractions were estimated from their Figure 4. Abbreviations: A1: primary auditory cortex; A7: Brodmann area 7; MT: middle temporal area; V1: primary visual cortex; V2: secondary visual cortex; 3: primary somatosensory cortex; 4: primary motor cortex.

The online version of this article includes the following figure supplement(s) for figure 3:

**Figure supplement 1.** Ferumoxytol-induced change in $R_2$ ($\Delta R_2$*) across equivolumetric layers (ELs).

and cytoarchitecture ($\tau = 0.73$, $p < 10^{-6}$). In the isocortex, the majority of the areas exhibited a distinctively high $\Delta R_2$* in EL4 (*Figure 5A, E*). The primary input layer, approximated as EL4a/b, exhibited systematically higher vascularity than in the primary output layer ($p < 10^{-25}$), approximated as EL5a/b. In contrast, the majority of the agranular and dysgranular areas (cluster3; *Figure 5*) exhibited weak laminar differentiation and a modest vascular density peak in EL1–2 (*Figure 5A*).

Given neurons and receptors collectively constitute approximately 75–80% of the brain's total energy budget (*Howarth et al., 2012*; *Hyder et al., 2013*), we next asked whether the regional variation in cerebrovascular network architecture (*Figure 1E*) is associated with heterogeneous cellular and receptor densities. To address this question, we applied linear regression model, utilizing quantitative neuron (*Collins et al., 2010*) and receptor density maps (*Froudist-Walsh et al., 2023*), to predict variation in $\Delta R_2$* (*Figure 6A, B, D*). This analysis revealed a positive correlation between CBV and neuron density in the middle cortical layers, where neuron density is typically highest, while revealing

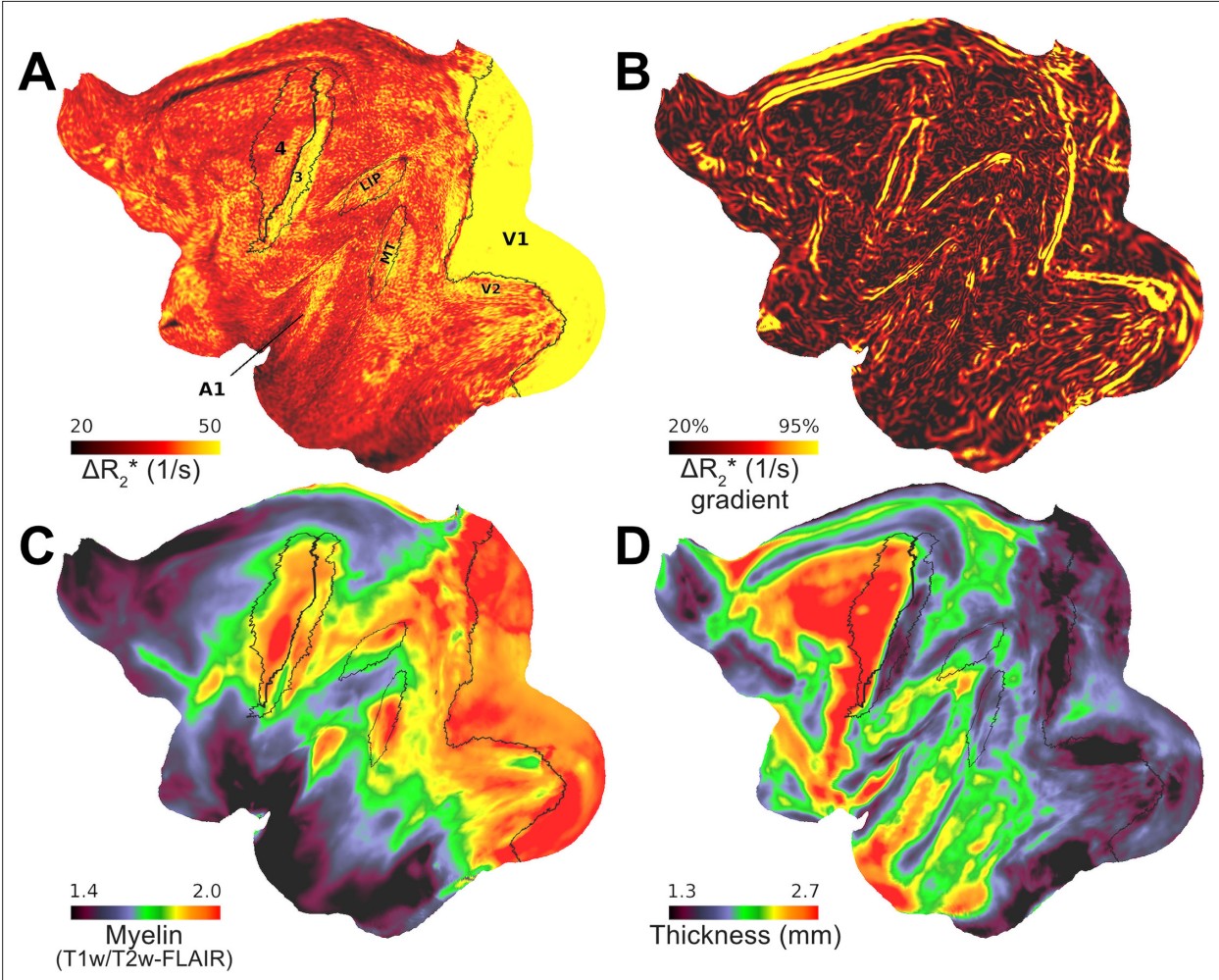

**Figure 4.** Variations in vascular network architecture reveal cortical area boundaries. (**A**) Ferumoxytol-induced change in transverse relaxation rate ($\Delta R_2$*) displayed at a representative equivolumetric layer 4b (EL4b) ($N = 4$). Overlaid black lines show exemplary M132 atlas area boundaries. (**B**) $\Delta R_2$* gradients co-align with exemplary areal boundaries. Red arrow indicates an artifact from inferior sagittal sinus. Average (**C**) mid-thickness weighted T1w/T2w-FLAIR myelin and (**D**) cortical thickness maps.

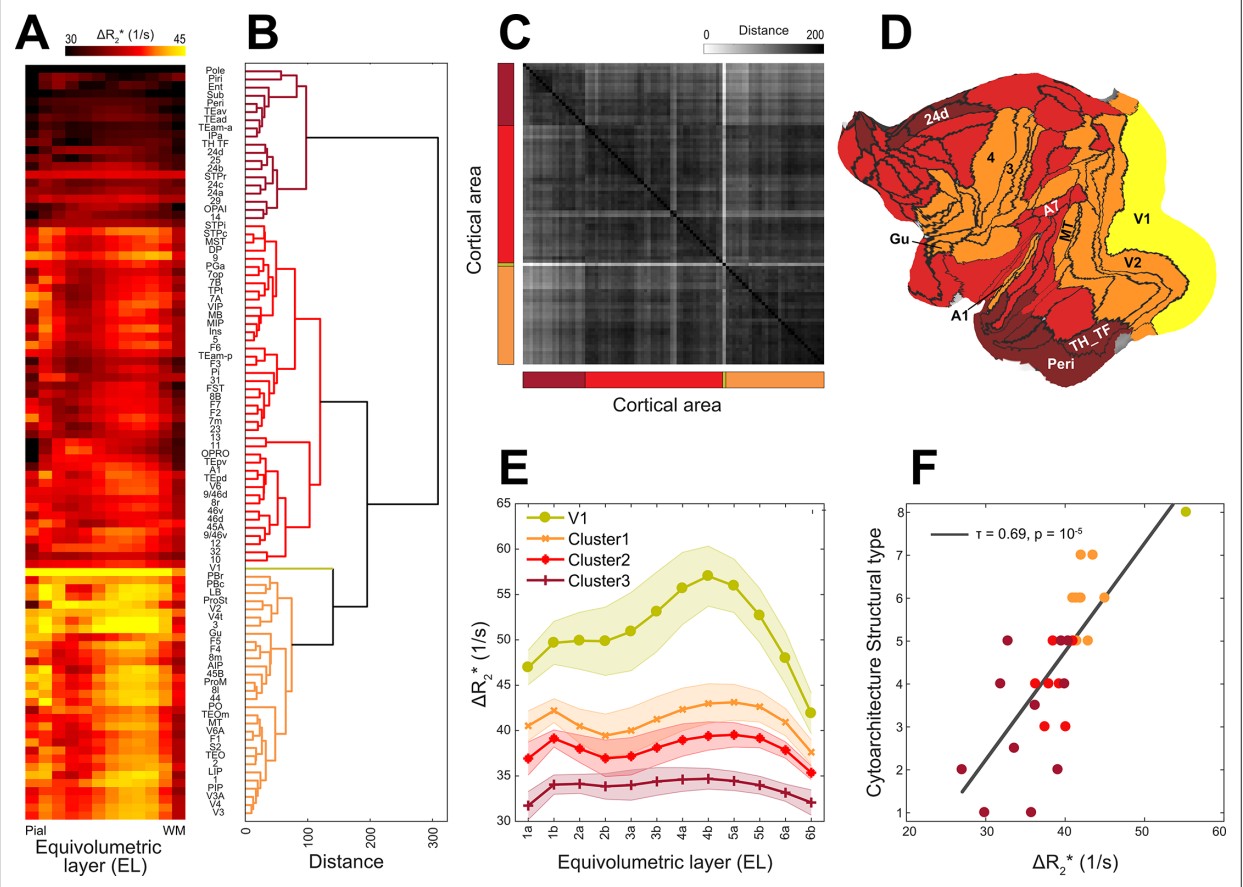

**Figure 5.** Hierarchical organization and principal types of cerebral vasculature. (**A**) Average $\Delta R_2^*$ equivolumetric layers (ELs) ascending from pial surface (left) to white matter surface (right) ($N = 4$; hemispheres = 8). Parcel order was sorted by (**B**) dendrogram determined using Wards' method. (**C**) Similarity matrix as estimated using Euclidean distance. (**D**) Clusters displayed on a cortical flat-map. (**E**) Average cluster profiles. Error bar indicates standard deviation across parcels within each cluster. (**F**) Cytoarchitectonic structural type co-vary with $\Delta R_2^*$.

a negative correlation with receptor density in the superficial layers where synaptic density is highest (*Figure 6F*). Additionally, we observed that baseline $R_2^*$ exhibited positive correlation with neuron density and negative correlation with receptor density (*Figure 6E*).

Because the Julich cortical area atlas covers only a section of the cerebral cortex, and the neuron density estimates are interpolated maps, we extended our analysis using the original Collins sample borders encompassing the entire cerebral cortex (*Figure 6—figure supplement 1A–C*). This analysis reaffirmed the positive correlation with $\Delta R_2^*$ (peak at EL2, $R = 0.80$, $p < 10^{-11}$) and baseline $R_2^*$ (peak at EL2a, $R = 0.86$, $p < 10^{-13}$), yielding linear coefficients of $\Delta R_2^* = 102 \times 10^3$ neurons/s and $R_2^* = 41 \times 10^3$ neurons/s (*Figure 6—figure supplement 1D–G*). This suggests that the sensitivity of quantitative layer $R_2^*$ MRI in detecting neuronal loss is relatively weak, and the introduction of the ferumoxytol contrast agent has the potential to enhance this sensitivity by a factor of 2.5.

Having established that vascular volume is associated with fundamental building units of cortical microcircuitry (*Figures 5F and 6E, F*), our subsequent inquiry aimed to explore connection with interneurons that govern the neuroenergetics of local neural networks (*Buzsáki et al., 2007*). By utilizing the interneuron densities mapped onto M132 atlas (*Burt et al., 2018*), we identified a positive correlation between $\Delta R_2^*$ and parvalbumin interneuron density (peak at EL5b, $R = 0.72$, $p < 10^{-6}$; Bonferroni corrected). In contrast, $\Delta R_2^*$ showed negative correlation with the density of calretinin-expressing slow-spiking interneurons which preferentially target distal dendrites (peak EL1b, $R = -0.61$, $p < 0.001$; Bonferroni corrected). In conjunction, we found that $\Delta R_2^*$ was also negatively correlated with dendritic tree size ($R = -0.46$, $p < 0.01$) (*Figure 6—figure supplement 2A, D and E*), the number of spines in L3 pyramidal cells ($R = -0.37$, $p = 0.06$), and also with $R_2^*$ ($R = -0.69$, $p < 0.001$) (*Figure 6—figure supplement 2B, D and F*; *Elston, 2002*; *Froudist-Walsh et al., 2023*). These findings establish the

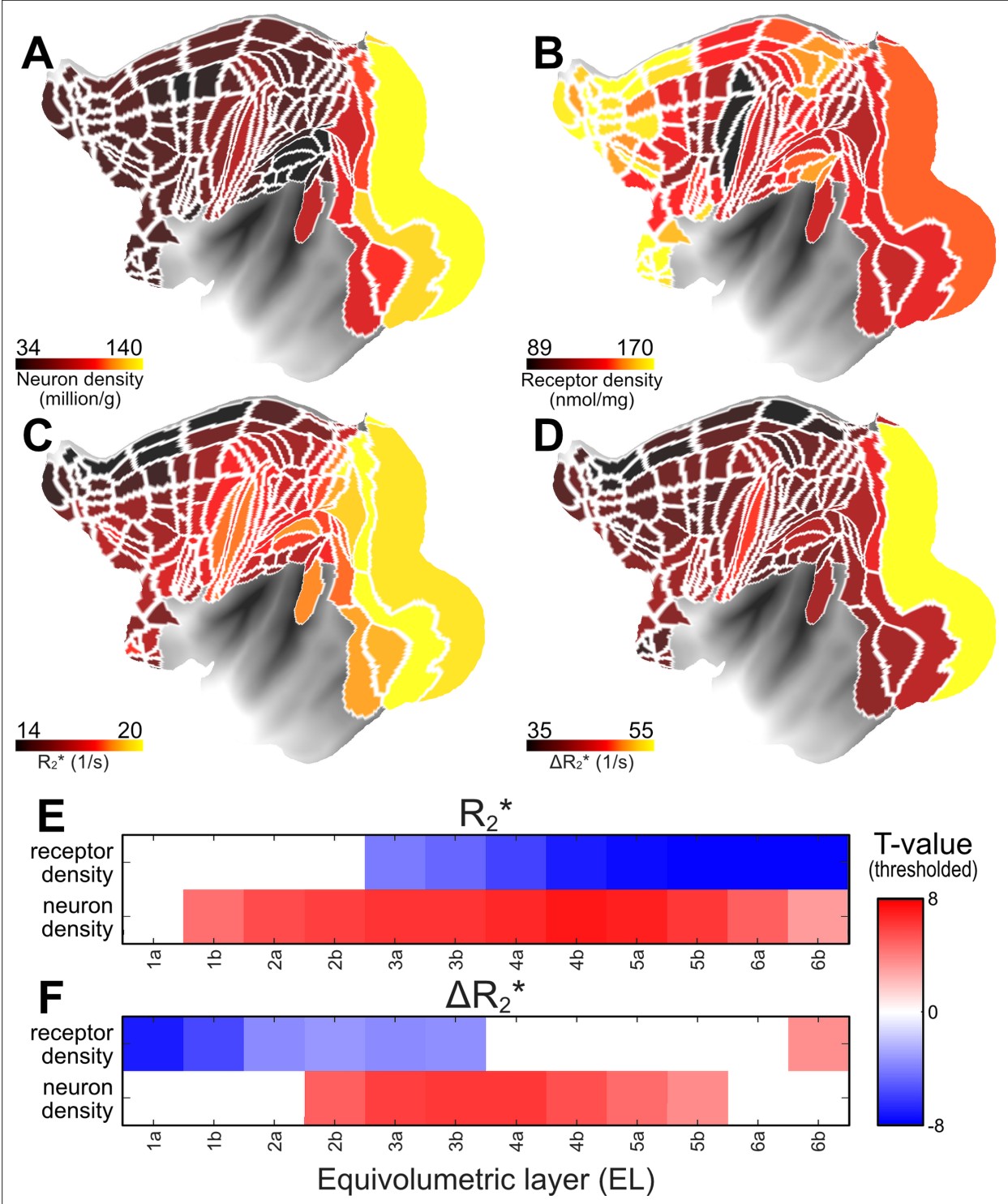

**Figure 6.** The anatomical underpinnings of the vascular network architecture. (**A**) Neuron (*Collins et al., 2010*), (**B**) total receptor density (*Froudist-Walsh et al., 2023*), and (**C, D**) $R_2^*$ and $\Delta R_2^*$ (current study). Multiple linear regression model was used to investigate the relationship between neuron and total receptor densities and (**E**) baseline $R_2^*$ and (**F**) $\Delta R_2^*$ across layers. *T*-values are threshold at significance level ($p < 0.05$, Bonferroni corrected).

The online version of this article includes the following figure supplement(s) for figure 6:

**Figure supplement 1.** Comparison with neuron density and baseline $R_2^*$ and $\Delta R_2^*$ in the entire cerebral cortex.

**Figure supplement 2.** Anatomical underpinnings of heterogeneous vascular density.

**Figure supplement 3.** Cerebrovascular volume varies along the cortical hierarchy.

intricate relationship between vascular density and the regulatory mechanisms governing diverse neural circuitry within the cerebral cortex.

## Discussion

We present a noninvasive methodology to evaluate layer variations in vascular network architecture in the primate cerebral cortex. The quantitative cortical layer thickness adjusted ferumoxytol-weighted MRI enables exploration of systematic variations in cortical energy supply architecture and vessel-frequency informed image acquisition enables benchmarking penetrating vessel density measures relative to the anatomical 'ground-truth'. These advances enabled us to unravel the systematic relation between vascularity and neurometabolic factors such as neuron and synaptic densities. Altogether, our study provides methodological and conceptual advancements in the field of cerebrovascular imaging.

### Methodological considerations—vessel-density informed MRI

To gain insights into the organization of cerebrovascular networks, it is important to critically sample the large irrigating arteries and draining veins while preserving adequate SNR in gray matter. While the pial vessels can be directly visualized using high-resolution time-of-flight MRI (***Bollmann et al., 2022***), and computed tomography (***Starosolski et al., 2015***), imaging of the dense vascularity within the large and highly convoluted primate gray matter presents other formidable challenges. Here, we used a combination of ferumoxytol contrast agent and laminar-resolution 3D GRE MRI to map cerebrovascular architecture in macaque monkeys. These methods allowed us to indirectly delineate large vessels and estimate translaminar variations in cortical microvasculature.

This methodology, however, has known limitations. First, gradient-echo imaging is more sensitized toward large pial vessels running along the cortical surface and large penetrating vessels, which could differentially bias the estimation of $\Delta R_2^*$ across cortical layers (***Figure 2A, B***; ***Boxerman et al., 1995***; ***Zhao et al., 2006***). Additionally, vessel orientation relative to the $B_0$ direction introduce strong layer-specific biases in quantitative $\Delta R_2^*$ measurements (***Figure 1—figure supplement 1C***; ***Lauwers et al., 2008***; ***Ogawa et al., 1993***; ***Viessmann et al., 2019***). To address these concerns, we conducted necessary corrections for $B_0$-orientation, obtained parcel median values and regressed linear-trend thereby mitigating the effect of undersampling large-caliber vessels across ELs (***Figure 2C***, ***Figure 1—figure supplement 1***). These analytical solutions yielded $\Delta R_2^*$ V1 translaminar profiles that more closely resembled capillary rather than large vessel volume profiles thus substantiating the validity of our methodology (***Figure 3F***; ***Weber et al., 2008***).

Ferumoxytol-weighted MRI of macaque cerebral cortex also enables the benchmarking of the methodological strengths and limitations to noninvasively measure vessel and vessel network length densities relative to the 'ground-truth'. In macaque V1, large vessel length density (threshold at 8 µm diameter) is 138 mm/mm$^3$ and mean area irrigated or drained by the vessels are about 0.26 and 0.4 mm$^2$ for arteries and veins, respectively (***Weber et al., 2008***). Combined, the total vessel density (=artery/0.26 + vein/0.40 mm$^2$) is 6.3 vessels/mm$^2$ (***Zheng et al., 1991***; ***Weber et al., 2008***), but see ***Adams et al., 2015***; ***Keller et al., 2011***. Based on the former literature estimates, we hypothesized that isotropic voxel of 0.23 mm (19 voxels/mm$^2$) may enable critical sampling of large vessels in accordance with sampling theorem (the sampling frequency equals to or is greater than twice the spatial frequency of the underlying anatomical detail in the image). In V1, we found an average vessel density of 2.2 ± 0.7 vessels/mm$^2$ (***Figure 2E***) which corresponds to ≈30% of the 'ground-truth' estimate (***Weber et al., 2008***). Using cortical thickness as a reference, we estimate that the vessel length density is ≈8 mm/mm$^3$ which corresponds to a modest ≈10% of the 'ground-truth' (***Weber et al., 2008***). The latter underestimate may be attributed to under-sampling of the branching arteriole and venule networks. Indeed, anatomical studies accounting for branching patterns have reported much higher vessel densities up to 30 vessels/mm$^2$ (***Keller et al., 2011***; ***Adams et al., 2015***). Further investigations are warranted, taking into account critical sampling frequencies associated with vessel branching patterns (***Duvernoy et al., 1981***), achieving higher SNR through ultra-high $B_0$ MRI (***Bolan et al., 2006***; ***Harel et al., 2010***; ***Kim et al., 2013***) and utilize high-resolution single-plane sequences and prospective motion correction schemes to accurately characterize regional vessel densities. Such advancements hold promise for improving vessel quantification, classifications for veins and arteries

and constructing detailed cortical surface maps of the vascular networks which may have diagnostic and neurosurgical utilities (*Figure 2A, B*; *Iadecola, 2013*; *Qi and Roper, 2021*; *Sweeney et al., 2018*).

## Sharp transitions in microvasculature are indicative of cortical area- and layer-specific energy requirements

These methodological advances enabled us to unveil variations in vascular density within the primate cerebral cortex. Primary sensory cortices, known for their high energy demands, exhibit distinctive vascularization patterns (*Figures 1C, D and 3B*). Notably, V1, area 3, auditory cortex, and also MT, all demonstrate elevated levels of cytochrome oxidase (CO) enzymatic activity compared to surrounding cortical regions (*Hackett et al., 1998*; *Horton, 1984*; *Huntley and Jones, 1991*; *Krubitzer et al., 2004*; *Matelli et al., 1985*; *Morel et al., 1993*; *Sincich et al., 2003*). CO staining often reveals sharp transitions historically employed to delineate cortical area boundaries and modular features of the cortex. Our results corroborate the over three-decade-old, yet previously untested, hypothesis that some inter-areal boundaries may be determined by their microcapillary density using contrast-agent-weighted MRI (*Zheng et al., 1991*). Dense vascularity in these areas, the sharp gradient-ridges observed between surrounding areas and co-alignment with existing areal atlases further support this hypothesis (*Figure 4A, B*).

Beyond the primary sensory areas, our observations extend to various smaller layer-specific vascular transitions (*Figure 5A, B*). Specifically, the lateral intraparietal area exhibits high CBV and gradient-ridges relative to ventral intraparietal area and associative Brodmann area 7. In EL5b, a strong gradient-ridge was observed distinguishing areas F4 and 44 from 45B and 8L/m. We also note weaker vascular transitions between areas such as 3a vs 3b, 5 vs anterior intraparietal area, supplementary motor cortex (SII) vs insula, A1 vs medial belt and 46v vs 46d. However, our ability to confidently determine these borders is constrained by the presence of large vessels, as well as potential surface placement errors, and validating these areal boundaries would benefit from utilizing multimodal approaches.

In their work, Zheng et al. also proposed that modular features of the cortex, characterized by greater vascular density in the V1 CO blobs (42%) than interblobs, could be delineated using contrast-agent-weighted MRI (*Zheng et al., 1991*). Such a large vascular density difference should be well-within our contrast-to-noise and spatial sampling limitations. However, our results do not support this hypothesis, as we do not observe a distinct vascular peak at the spatial frequency of CO blobs (≈2.2 1/mm; *Figure 2D*). Our results align with anatomical studies challenging the existence of high capillary density in the V1 blobs (*Keller et al., 2011*; *Adams et al., 2015*).

The variation in vascular volume also has implications for statistical power in fMRI. For example, the pallidum exhibits the lowest vascular volume (*Figure 1C*) while in V1 EL4, the vascular volume is approximately twice as high (*Figure 3D*). According to the classical single-tissue compartment model, CNR is optimized when TE is matched with $T_2^*$. Consequently, there is no single optimal TE for CBV weighted fMRI. Multi-echo EPI acquisition (*Kuroiwa et al., 2014*; *Poser and Norris, 2009*) may provide a more balanced comparison for statistical power across different brain regions and cortical layers.

## The vascular network architecture is intricately connected to the neuroanatomical organization within cerebral cortex

Given the fivefold variability in neuron density across the cortex (*Collins et al., 2010*), one might expect that there is a corresponding variation in cerebral blood flow (CBF) and CBV (*Figure 6*; *Tsai et al., 2009*). In the cerebral cortex, neurons account for a significant portion (≈80–90%) of energy demand, with most of this energy allocated to signaling (≈80%) and maintaining membrane resting potentials (≈20%) (*Attwell and Laughlin, 2001*; *Howarth et al., 2012*). Since firing frequency is modulatory and the neural networks utilize distributed coding, the maintenance of resting-state membrane potential determines the minimal energy budget and the lower-limit for cerebral perfusion. Based on neuronal variability and energy dedicated to maintaining surface potential, this suggest an approximate (4 × 20% ≈) 80% variation in CBF and a resultant 25% variation in CBV across the cortex, in line with Grubbs' law (CBV = 0.80 × $CBF^{0.38}$) (*Grubb et al., 1974*). In the cerebellar cortex, neuron density is higher, and the resting potentials are thought to account for more than 50% of energy usage (*Howarth et al., 2012*), aligning with its higher vascular volume compared to the cerebral cortex (*Figure 1F*). However, this is a simplified estimation, and a more comprehensive assessment would

need to account for an aggregate of biophysical factors such as neuron types, neuron membrane surface area, firing rates, dendritic and synaptic densities (*Figure 6F, G*), neurotransmitter recycling, and other cell types (*Kageyama and Wong-Riley, 1982*; *Elston and Rosa, 1997*; *Perge et al., 2009*; *Harris and Attwell, 2012*). Indeed, the majority of the mitochondria reside in the dendrites and synaptic transmission is widely acknowledged to drive the majority of the energy consumption and blood flow (*Wong-Riley, 1989*; *Attwell et al., 2010*).

Extrapolating cortical $\Delta R_2^*$ to zero neuron density results in a large intercept (~35 1/s), corresponding to 60% of the maximum cortical CBV (57 1/s; *Figure 6—figure supplement 1F*). This supports the view that most of the energy consumption occurs in the neuropil—comprising dendrites, synapses, and axons—which accounts for ~80–90% of cortical gray matter volume, whereas neuronal somata constitute only ~10–20% (*Wong-Riley, 1989*). Although neuronal cell bodies exhibit higher CO activity per unit volume due to their dense mitochondrial content, these results suggest their overall contribution to the total CBV per mm³ tissue remains lower than that of the neuropil, given the latter's substantially larger volume fraction in cortical tissue.

Contrary to our initial expectations, we observed a relatively smaller CBV in regions and layers with high receptor density (*Figure 6B, D, F*). This relationship extends to other factors, such as number of spines (putative excitatory inputs) and dendrite tree size across the entire cerebral cortex (*Figure 6—figure supplement 2*; *Froudist-Walsh et al., 2023*; *Elston, 2007*). These results align with the work of Weber et al., who reported a similar negative correlation between vascular length density and synaptic density, as well as a positive correlation with neuron density in macaque V1 across cortical layers (*Weber et al., 2008*). This relation is also compatible with the opposing relation between CBV and GABAergic (GABA, γ-aminobutyric acid) interneuron subtypes: parvalbumin-expressing fast-firing interneurons target perisomatic parts of pyramidal neurons, whereas calretinin-expressing slow-firing interneurons target distal-dendrites. The interneurons are also well positioned to play an important role in integrating activity of large numbers of excitatory principal cells and translating this into neuro-vascular regulation of local microcirculation via subcortical pathways (*Cauli et al., 2004*). Another perspective on our results considers that a relative smaller faction of synapses may be simultaneously active in larger neurons, and neurons with large dendrites may exhibit lower excitability, supporting pattern separation necessary for higher-level functions (*Chavlis et al., 2017*; *Hawkins and Ahmad, 2016*). To comprehensively understand the factors contributing to the vascular organization of the brain, experimental disentanglement through multivariate analysis of laminar cell types and receptor densities is needed (*Hayashi et al., 2021*; *Froudist-Walsh et al., 2023*). Moreover, employing more advanced statistical modeling, including considerations for synapse-neuron interactions, may be important for refined evaluations.

Another key finding of this study was the strong correlation between baseline $R_2^*$ and neuron density (*Figure 6—figure supplement 1D, E*). While $R_2^*$ is well known to be influenced by iron, myelin, and deoxyhemoglobin densities, this correlation peaks in the superficial layers (*Figure 6—figure supplement 1E*), suggesting a link to CO activity and the accumulation of deoxygenated venous blood draining from all cortical layers toward the pial network. Notably, the absolute range of superficial $R_2^*$ values (max − min ≈ 6 s⁻¹; *Figure 6—figure supplement 1D*) is approximately 12–30 times larger than the $\Delta R_2^*$ observed during task-based BOLD fMRI at 3T (0.2–0.5 1/s) (*Yablonskiy and Haacke, 1994*). Since venous oxygenation is around 60% and task-induced changes in blood flow account for only 5–10% of the brain's resting blood flow (*Raichle and Mintun, 2006*), these results suggest that superficial $R_2^*$ (*Figure 1D*) may serve as a more accurate proxy for total deoxyhemoglobin content (and thus total oxygen consumption), which scales with the neuron density of the underlying cortical gray matter. Importantly, superficial layers may also provide a more specific measure of deoxyhemoglobin, as they are less influenced by myelin and iron, which are more concentrated in deeper cortical layers. Additionally, smaller but direct contributors, such as mitochondrial CO density—an iron-dependent factor—may also play a role in this relationship.

Additionally, our investigation also uncovered an overlap between vascular volume and myelin (*Figure 3A, C*). Myelin may indirectly contribute to increased energy consumption in the cortical gray matter by facilitating higher frequency firing in comparison to unmyelinated axons (*Perge et al., 2012*; *Saab et al., 2016*). A large fraction of cortical myelin enwraps the axons of parvalbumin-expressing fast-spiking interneurons (*Stedehouder et al., 2017*). Although interneurons represent a minority of the neuronal population (10–15%), the parvalbumin-expressing interneurons' ability to

sustain high-frequency gamma oscillations (30–100 Hz) may match the sparse firing of the majority principal cells (**Buzsáki et al., 2007**). Indeed, parvalbumin-expressing interneurons exhibit higher mitochondrial volume compared to other cells in the brain and a threefold higher CO activity than principal neurons (**Gulyás et al., 2006**; **Kageyama and Wong-Riley, 1982**; **Kann et al., 2014**; **Nie and Wong-Riley, 1995**). Thus, the high metabolic load of parvalbumin-expressing interneurons makes them potentially vulnerable to failures in the vascular network due to aging, Alzheimer's disease as well as stroke (**Kann, 2016**).

Given the distinct pre- and postsynaptic metabolic requirements, heterogeneous translaminar vascularization may also indicate distinct cortical layers each characterized by anatomically and physiologically distinct feedforward and feedback pathways (**Bastos et al., 2015**; **Borowsky and Collins, 1989**; **Kageyama and Wong-Riley, 1982**; **Takahata, 2016**). A prime example is V1 where the primary input layer 4 (L4) has more dense vascularization and 50% higher CO activity in comparison with primary output L5 (**Figure 3F**; **Keller et al., 2011**; **Livingstone and Hubel, 1982**). This elevated energy demand in L4 may arise, in part, from spontaneous supra-threshold gamma-frequency oscillations between the retina→lateral geniculate nucleus→L4 (**Castelo-Branco et al., 1998**) along with recurrent amplification of local and distant inputs (**Douglas and Martin, 2007**; **Shu et al., 2003**). When viewed in terms of information flow, CBV appear to decrease along the canonical circuit pathway (e.g., L4→L2/3→L5) in the primary visual cortex (**Douglas and Martin, 2007**) and as one ascends the hierarchy (e.g., V1→V2→V3&4→MT→7A) from primary sensory areas (**Figure 3F**, **Figure 6—figure supplement 3**; **Felleman and Van Essen, 1991**; **Markov et al., 2014a**). A similar pattern is observed in the auditory hierarchy, where the inferior colliculus, an early processing hub, exhibits the highest vascular volume, followed by a gradual reduction along cortical auditory 'where' and 'what' pathways (**Figures 1F and 3B**). In the agranular and dysgranular regions, characterized by a lack of distinct L4, the translaminar CBV profiles did not exhibit a distinct peak at around EL4 nor signatures of canonical circuitry (**Figure 5**). These results demonstrate a greater allocation of the energy budget to early stages of feedforward processing in primary cortical areas characterized by strong sensory inputs, as opposed to higher-level cortical areas characterized by high synaptic densities supporting cognitive and behavioral functions (**Figure 6B**; **Yokoyama et al., 2021**).

The anatomical uniformity of the primate neocortex is thought to reflect extensive replication of a few specialized microcircuits varying along the brain's hierarchical organization (**Douglas and Martin, 2007**; **Hilgetag et al., 2016**; **Markram et al., 2004**). Analogous to cortical circuitry, our study reveals the large-scale replication of translaminar vascular network motifs in primates (**Figure 4B–D**). Since the cerebrovascular system evolved to support the high energy demands of neural information processing, this raises questions about whether the large-scale replication of anatomical and vascular circuits is evolutionarily coupled (**Carmeliet and Tessier-Lavigne, 2005**). In mice, comprehensive analysis of the cerebrovascular system has revealed distinct translaminar types between sensory and motor-integrative areas (**Kirst et al., 2020**). In macaque, we found the strongest distinction between isocortex and allocortex and its adjacent regions (**Figure 6B**). The species difference may reflect evolutionary expansion and emergence of new cortical layers. For instance, in mice the primary somatosensory cortex exhibits highest vascularization (**Kirst et al., 2020**) whereas our results show that in macaque the highest vascularization is in the V1 (**Figure 5A, E**; **Duvernoy et al., 1981**). According to the theory that sensory systems, behavior, and habitat choice are all influenced by evolutionary processes (**Endler, 1992**; **Ikeda et al., 2023**), this may reflect an evolutionary adaptation to an environment in which the visual landscape is an ecologically more important sensory domain in primates.

## Materials and methods

### Data acquisition

Experiments were performed using a 3T MRI scanner (MAGNETOM Prisma, Siemens, Erlangen, Germany) equipped with 80 mT/m gradients (XR 80/200 gradient system with slew rate 200 T/m/s), a 2-channel $B_1$ transmit array (TimTX TrueForm) and a custom-made 24-channel coil for the macaque brain (**Autio et al., 2020**). The animal experiments were conducted in accordance with the institutional guidelines for animal experiments, and animals were maintained and handled in accordance with the policies for the conduct of animal experiments in research institution (MEXT, Japan, Tokyo) and the Guide for the Care and Use of Laboratory Animals of the Institute of Laboratory Animal

Resources (ILAR; Washington, DC, USA). All animal procedures were approved by the Animal Care and Use Committee of the Kobe Institute of RIKEN (MA2008-03-11).

## Anesthesia protocol

Macaque monkeys (*Macaca mulatta*, weight range 7.4–8.4 kg, age range 4–6 years, $N = 4$) were initially sedated with intramuscular injection of atropine sulfate (20 µg/kg), dexmedetomidine (4.5 µg/kg), and ketamine (6 mg/kg). A catheter was inserted into the caudal artery for blood-gas sampling, and endotracheal intubation was performed for steady controlled ventilation using an anesthetic ventilator (Cato, Drager, Germany). End-tidal carbon dioxide was monitored and used to adjust ventilation rate (0.2–0.3 Hz) and end-tidal volume. After the animal was fixed in an animal holder, anesthesia was maintained using 1.0% isoflurane via a calibrated vaporizer with a mixture of air 0.75 l/min and $O_2$ 0.1 l/min. Animals were warmed with a blanket and water circulation bed and their rectal temperature (1030, SA Instruments Inc, NY, USA), peripheral oxygen saturation and heart rate (7500FO, NONIN Medical Inc, MN, USA) were monitored throughout experiments.

## Structural acquisition protocol

T1w images were acquired using a 3D Magnetization Prepared Rapid Acquisition Gradient Echo (MPRAGE) sequence (0.5 mm isotropic, FOV 128 × 128 × 112 mm, matrix 256 × 256, slices per slab 224, coronal orientation, readout direction of inferior (I) to superior (S), phase oversampling 15%, averages 3, TR 2200 ms, TE 2.2 ms, TI 900 ms, flip-angle 8.3°, bandwidth 270 Hz/pixel, no fat suppression, GRAPPA 2, turbo factor 224 and pre-scan normalization). T2w images were acquired using a Sampling Perfection with Application optimized Contrast using different angle Evolutions (SPACE) sequence (0.5 mm isotropic, FOV 128 × 128 × 112 mm, matrix 256 × 256, slice per slab 224, coronal orientation, readout direction I to S, phase oversampling 15%, TR 3200 ms, TE 562 ms, bandwidth 723 Hz/pixel, no fat suppression, GRAPPA 2, turbo factor 314, echo train length 1201ms and pre-scan normalization) (*Autio et al., 2021*; *Autio et al., 2020*).

In a separate imaging session, additional high-resolution structural images were acquired (*Autio et al., 2024*). T1w images were acquired using a 3D Magnetization Prepared Rapid Acquisition Gradient Echo (MPRAGE) sequence (0.32 mm isotropic, FOV 123 × 123 × 123 mm, matrix 384 × 384, slices per slab 256, sagittal orientation, readout direction FH, averages 12–15, TR 2200 ms, TE 3 ms, TI 900 ms, flip-angle 8°, bandwidth 200 Hz/pixel, no fat suppression, GRAPPA 2, reference lines PE 32, turbo factor 224, averages 12–15, and pre-scan normalization). T2w images were acquired using a Sampling Perfection with Application optimized Contrast using different angle Evolutions and Fluid-Attenuated Inversion Recovery (SPACE-FLAIR) sequence (0.32 mm isotropic, FOV 123 × 123 × 123 mm, matrix 384 × 384, slice per slab 256, sagittal orientation, readout direction FH, TR 5000 ms, TE 397 ms, TI 1800 ms, bandwidth 420 Hz/pixel, no fat suppression, GRAPPA 2, reference lines PE 32, turbo factor 188, echo train duration 933 ms, averages 6–7 and pre-scan normalization). The total acquisition time for structural scans was ≈3 hr.

## Quantitative transverse relaxation rate acquisition protocol

Data was acquired before and after (12 mg/kg) the intravascular ferumoxytol (Feraheme, ferumoxytol AMAG Pharmaceuticals Inc, Waltham, MA, USA) injection using gradient- and RF-spoiled 3D multi-echo gradient-echo acquisition (0.32 mm isotropic, FOV 103 × 103 × 82 mm, matrix 320 × 320, slices per slab 256, sagittal orientation, bipolar read-out mode, elliptical scanning, no partial Fourier, ten equidistant TEs, first TE (TE1) = 3.4 ms, time between echoes (ΔTE) 2.4 ms, TR 33 ms, FA 13° (corresponding to Ernst angle of gray matter; median $T_1$ = 1370 ms), bandwidth 500 Hz/pixel (fat-water shift one voxel), scan duration 20 min, GRAPPA 2, reference lines 32, and pre-scan normalization). The total acquisition time before and after ferumoxytol injection were 40 and 100 min, respectively.

## Vessel-density informed data acquisition protocol

To investigate the periodicity of the penetrating large vessel network, we performed auxiliary ferumoxytol-weighted experiments using image resolution adjusted to satisfy critical (spatial) sampling frequency (14 voxels/mm² ≈0.26 mm isotropic) of intra-cortical vessels (7 vessels/mm²; *Weber et al., 2008*). The original gradient- and RF-spoiled 3D multi-echo gradient-echo product sequence, however, did not allow sufficient matrix size to satisfy the critical sampling frequency of penetrating vessels. To

achieve the target resolution, the sequence was customized by easing the matrix size limitations (by Y.U.). Using the customized sequence, we performed experiments at 0.25 ($N = 1$) and 0.23 mm ($N = 2$) isotropic spatial resolution. Scan #1: (FOV 104 × 104 × 80 mm, matrix 416 × 416, slices per slab 320, sagittal orientation, bipolar read-out mode, elliptical scanning, no partial Fourier, three TEs 6, 10, and 14 ms, TR 22 ms, FA 11°, bandwidth 260 Hz/pixel (fat-water shift 1.6 voxels), scan duration 21 min, GRAPPA 2, reference lines 32 and pre-scan normalization). Scans #2–3: (FOV 103 × 103 × 81 mm, matrix 448 × 448, slices per slab 352, sagittal orientation, bipolar read-out mode, elliptical scanning, no partial Fourier, three TEs 5, 9, and 13 ms, TR 23 ms, FA 11°, bandwidth 340 Hz/pixel (fat-water shift 1.2 voxels), scan duration 25 min, GRAPPA 2, reference lines 32, and pre-scan normalization). The total acquisition time was 150 min.

## Data analysis

Data analysis utilized a version of the HCP pipelines customized specifically for use with non-human primates (https://github.com/Washington-University/NHPPipelines, *Brown et al., 2021*; *Autio et al., 2020*; *Glasser et al., 2013*; *Hayashi et al., 2021*).

## Structural image processing

PreFreeSurfer pipeline registered structural T1w and T2w images into an anterior–posterior commissural (AC–PC) alignment using a rigid body transformation, non-brain structures were removed, T2w and T1w images were aligned using boundary based registration (*Greve and Fischl, 2009*), and corrected for signal intensity inhomogeneity using $B_1$-bias field estimate (*Glasser et al., 2013*). Next, data was transformed into a standard macaque atlas by 12-parameter affine and nonlinear volume registration using FLIRT and FNIRT FSL tools (*Jenkinson et al., 2002*).

FreeSurferNHP pipeline was used to reconstruct the cortical surfaces using FreeSurfer v6.0.0-HCP. This process included conversion of data in native AC–PC space to a 'fake' space with 1 mm isotropic resolution in volume with a matrix of 256 in all directions, intensity correction, segmentation of the brain into cortex and subcortical structures, reconstruction of the white and pial surfaces and estimation of cortical folding maps and thickness. The intensity correction was performed using FMRIB's Automated Segmentation Tool (FAST) (*Zhang et al., 2001*). The white matter segmentation was fine-tuned by filling a white matter skeleton to accurately estimate white surface around the blade-like thin white matter particularly in the anterior temporal and occipital lobe (*Autio et al., 2020*). After the white surface was estimated, the pial surface was initially estimated by using intensity normalized T1w image and then estimated using the T2w image to help exclude dura (*Glasser et al., 2013*).

The PostFreeSurfer pipeline transformed anatomical volumes and cortical surfaces into the Yerkes19 standard space, performed surface registration using folding information via MSMSulc (*Robinson et al., 2018*; *Robinson et al., 2014*), generated mid-thickness, inflated and very inflated surfaces, as well as the myelin map from the T1w/T2w ratio on the mid-thickness surface. The volume to surface mapping of the T1w/T2w ratio was carried out using a 'myelin-style' mapping (*Glasser and Van Essen, 2011*), in which a cortical ribbon mask and a metric of cortical thickness were used, weighting voxels closer to the midthickness surface. Voxel weighting was done with a Gaussian kernel of 2 mm FWHM, corresponding to the mean cortical thickness of macaque. For quality control, the myelin maps were visualized and potential FreeSurfer errors in pial or WM surface placement were identified. The errors were manually corrected by editing wm.mgz and by repositioning and smoothing the surfaces using FreeSurfer 7.1, the curvature, thickness, and surface area on each vertex were recalculated, and then PostFreeSurfer pipeline was applied again and T1w/T2w was visually inspected for quality control.

Twelve cortical laminar surfaces were generated based on equivolume model (*Autio et al., 2024*; *Van Essen and Maunsell, 1980*) using the Workbench command '-surface-cortex-layer' and the native pial and white surface meshes in subject's AC–PC space. Throughout the text, the ELs are referred to as EL1a (adjacent to the pial surface), EL1b, EL2a, EL2b,..., and EL6b (adjacent to the white matter surface). This nomenclature is intended to ease but also distinguish comparison between anatomically determined cortical layers which vary in thickness. Anatomical layers are referred to using roman numerals (e.g., Ia, Ib, Ic, IIa,…, and VIb). To assess the vascularity on the white matter and pial surfaces, additional layers were generated underneath and just above the gray matter in the superficial white matter and pial surface, respectively. Surface models and data were resampled to a high-resolution 164k mesh (per hemisphere).

## Quantitative multi-echo gradient-echo data processing

The original 3D multi-echo gradient-echo images were upsampled to 0.25 or 0.15 mm spatial resolution for the data with spatial resolution 0.32 or 0.23 and 0.26, respectively; and transformed using cubic-spline to the subject's AC–PC space using a rigid body transformation. Pre- and post-ferumoxytol runs (2 and 6, respectively) were averaged and $R_2$*-fitting procedure was performed on multi-TE images with ordinary least squares method in the in vivo histology using MRI (hMRI) Toolbox (*Tabelow et al., 2019*). The baseline (pre-ferumoxytol) $R_2$* was subtracted from the post-ferumoxytol $R_2$* maps to calculate ferumoxytol-induced change in $\Delta R_2$* (*Boxerman et al., 1995*). Subcortical region-of-interests (thalamus, striatum, cerebellum, hippocampus, inferior colliculus, and corpus callosum) were manually drawn while avoiding large vessels using T1w image as a reference. The quantitative $R_2$* and $\Delta R_2$* maps were mapped in the 12 native laminar mesh surfaces using the Workbench command '-volume-to-surface-mapping' using a ribbon-constrained algorithm. MSMSulc surface registration was applied, the data was resampled to Mac25Rhesus reference sulcus template using ADAP_BARY_AREA with vertex area correction, and left and right hemispheres were combined into a CIFTI file.

Since large-caliber pial vessels run along cortical surface, large penetrating vessels are mainly oriented normal to the cortical surface and the capillary network may be orientated more random to the cortical surface (*Ji et al., 2021*; *Reina-De La Torre et al., 1998*), the orientation of the cerebral cortex relative to the direction of static magnetic field (B$_0$) may bias the assessment of $R_2$* and $\Delta R_2$* (*Bolan et al., 2006*; *Lee et al., 2011*; *Ogawa et al., 1993*; *Viessmann et al., 2019*; *Yablonskiy and Haacke, 1994*). Because the brain $R_2$* measures are primarily determined by extravascular MR signal, we may assume that

$$R_2^* \propto \cos^2\left(\theta_{B_0}\right) \tag{1}$$

where $\Theta$ is the angle between normal of the cortex relative to the direction of B$_0$. Each vertex $\Theta$ was determined in the subject's original MRI space. The (Pearson's) correlation coefficient between *Equation 1* and with $R_2$* and $\Delta R_2$* was estimated in each EL. To remove orientation bias, $\cos^2\left(\theta_{B_0}\right)$ in *Equation 1* was regressed out from each laminar $R_2$* and $\Delta R_2$* surface map.

To examine repetitive patterns in the vascularity, the $R_2$* and $\Delta R_2$* laminar profiles were parcellated using the M132 Lyon Macaque brain atlas (*Markov et al., 2014a*). Because some of the M132 atlas cortical parcels exhibited a degree of laminar inhomogeneity due to artifacts (e.g., areas adjacent to major sinuses, large vessels penetrating to white matter and FreeSurfer errors in surface placement), median values were assigned to each parcel in each EL. The effect of blood accumulation in large feeding arteries and draining veins toward the superficial layers was estimated using linear model and regressed out from the parcellated $\Delta R_2$* maps. Subjects (including a single test–retest dataset), ELs and hemispheres were combined ($5 \times 12 \times 2 = 120$) and hierarchical clustering was applied to parcels using Ward's method. A dendrogram was used to determine the number of clusters.

To explore sharp transitions in cortical vascularization, each B$_0$ orientation corrected $\Delta R_2$* EL was smoothed using a factor of 1.2 mm. The smoothing factor was twice the average distance of draining veins (*Weber et al., 2008*). Then, gradient-ridges were calculated using maps using wb_command -cifti-gradient for each EL. The resulting gradients were then cross-referenced with potential Free-Surfer surface error displacements and when required manual corrections (wm.mgz and using reposition surface in the FreeView 3.0) were performed to the FreeSurfer segmentations.

## Vessel detection

To improve contrast-to-noise ratio for vessel detection, multi-echo gradient-echo images were aligned in the native AC–PC space and averaged across runs. Vessels were identified using the Frangi 'vessel-ness' filter which enhances the vessel/ridge-like structures in 3D image using hessian eigenvalues (*Avadiappan et al., 2020*; *Frangi et al., 1998*). Volume images in the native AC–PC space were also non-linearly transformed into a standard 'SpecMac25Rhesus' atlas (*Hayashi et al., 2021*).

To facilitate visualization of the pial vessel network, low-frequency fluctuations were removed by subtracting extensively smoothed versions of the post-ferumoxytol TE-averaged EL surface maps. To detect vessels running parallel to the cortical surface, continuous signal dropouts were clustered along ELs using wb_command -cifti-find-clusters with a criterion of a 0.5-mm$^2$ minimum cluster area and visually determined intensity threshold.

To enable surface detection of penetrating vessels, an ultra high-resolution 656k cortical surface mesh was generated using wb_command -surface-create-sphere resulting in an average 0.022 vertex surface area, approximately half the isotropic 0.23 mm voxel (face) surface area ($0.053 \text{ mm}^2$). Then, TE-averaged multi-echo gradient-echo images and Frangi-filtered vessel images were mapped to 12 native mesh laminar surfaces in the subject's physical space. To detect penetrating vessels oriented perpendicular to the laminar surfaces, localized signal drop-outs were visualized using wb_command -cifti-gradient and their central locations were identified by detecting the local minima using wb_command -find-extrema.

The number of vessels in V1 was estimated using the M132 parcellation as a reference (*Markov et al., 2014a*). In the native space, the surface area of each vertex was determined using wb_command -surface-vertex-areas. Then, the surface area map was transferred into atlas space and the area of V1 was determined using M132 areal atlas. The V1 vessel density was determined by dividing the number of vessels (by a conservative estimate using Frangi-filtering and a liberal estimate using local minima in gradient) by surface area. These MRI estimates were then compared with histological large vessel densities in the V1 (*Zheng et al., 1991*; *Weber et al., 2008*).

To determine the periodicity of the cortical arterio-venous networks, non-uniformly sampled Lomb–Scargle geodesic periodogram analysis (Matlab Signal Processing Toolbox, The MathWorks Inc, US) was performed on the spatially low-frequency filtered 12 native mesh ELs in the subject's physical space. The analysis was limited to the closest 2000 geodesic vertices within 20 mm geodesic distance of manually selected vertices in V1. Geodesic distance between vertices was calculated with wb_command -surface-geodesic-distance in each EL. Periodograms were binarized with an equidistant interval (=0.05 1/mm) up to 10 1/mm, and then 95% confidence interval of the mean of magnitude was estimated from bootstrap and compared across cortical laminae.

## Comparison with histological datasets

In V1, translaminar $\Delta R_2^*$ was compared with CO activity, capillary and large vessel volume fractions (*Weber et al., 2008*). These ground-truth measures were estimated from Weber et al. (*Figure 4*). Each measure was peak normalized, so that values ranged between 0 and 1, to compare different contrasts across the cortical layers.

To investigate the correspondence between regional variation in cerebrovascular volume and heterogeneous neuron density, we used the 42 Vanderbilt tissue sections covering the entire macaque cerebral cortex (*Collins et al., 2010*) available from the Brain Analysis Library of Spatial Maps and Atlases (BALSA) database (*Froudist-Walsh et al., 2023*; *Froudist-Walsh et al., 2021*; *Van Essen et al., 2017*). These sections were processed using the isotropic fractionator method to estimate neuron densities (*Collins et al., 2010*). The sections were used to parcellate $R_2^*$ and $\Delta R_2^*$ and these were then compared with neuron density using Pearson's correlation coefficient. To compare neuron and total receptor densities with $R_2^*$ and $\Delta R_2^*$, we also applied the Julich Macaque Brain Atlas for parcellation (*Froudist-Walsh et al., 2023*). The parcellated neuron and total receptor densities were used in linear regression model to predict $R_2^*$ and $\Delta R_2^*$ across ELs and the resulting *T*-values were then threshold at significance level ($p < 0.05$, Bonferroni-corrected).

Relation between cerebrovascular volume and parvalbumin and calretinin positive interneurons, collated from multiple studies and ascribed to M132 macaque atlas *Burt et al., 2018*; *Condé et al., 1994*; *Gabbott and Bacon, 1996*; *Kondo et al., 1999*, were compared across ELs using Pearson's correlation coefficient. The resulting p-values were Bonferroni-corrected for the number of layers and contrasts.

The number of dendritic spines (putative excitatory inputs) and dendrite tree length size were also obtained from BALSA (*Elston, 2007*; *Froudist-Walsh et al., 2023*; *Froudist-Walsh et al., 2021*). The region-of-interests, described in *Elston, 2007* and plotted on the Yerkes surface by Froudist-Walsh et al., were used to obtain median value of $R_2^*$ and $\Delta R_2^*$ and these were then compared with the number of dendritic spines and dendrite tree length size using Pearson's correlation coefficient.

## Acknowledgements

The authors appreciate discussions and technical contributions from Akiko Uematsu, Timothy S Coalson, Katsutoshi Murata, and Reiko Kobayashi. This research is partially supported by JSPS KAKENHI Grant Number (JP20K15945, JAA), by the program for Brain/MINDS and Brain/MINDS-beyond from Japan

Agency for Medical Research and development, AMED (JP18dm037006, JP23wm0625001, TH) and by NIH R01MH60974 (DCVE, MFG). The authors have no conflicts of interest to declare.

## Additional information

### Funding

| Funder | Grant reference number | Author |
|---|---|---|
| Japan Society for the Promotion of Science | JP20K15945 | Joonas A Autio |
| Japan Agency for Medical Research and Development | JP18dm037006 | Takuya Hayashi |
| National Institutes of Health | R01MH60974 | Matthew F Glasser David C van Essen |
| Japan Agency for Medical Research and Development | JP23wm0625001 | Takuya Hayashi |

The funders had no role in study design, data collection, and interpretation, or the decision to submit the work for publication.

### Author contributions

Joonas A Autio, Conceptualization, Resources, Data curation, Software, Formal analysis, Funding acquisition, Validation, Investigation, Visualization, Methodology, Writing – original draft, Project administration, Writing – review and editing; Ikko Kimura, Software, Formal analysis, Visualization, Writing – review and editing; Takayuki Ose, Yuki Matsumoto, Masahiro Ohno, Investigation; Yuta Urushibata, Resources, Methodology; Takuro Ikeda, Data curation, Formal analysis, Visualization; Matthew F Glasser, Writing – review and editing; David C van Essen, Resources, Writing – review and editing; Takuya Hayashi, Resources, Funding acquisition, Writing – review and editing

### Author ORCIDs

Joonas A Autio ⓘ https://orcid.org/0000-0002-2232-9259
Takuya Hayashi ⓘ https://orcid.org/0000-0001-7639-0197

### Ethics

The animal experiments were conducted in accordance with the institutional guidelines for animal experiments, and animals were maintained and handled in accordance with the policies for the conduct of animal experiments in research institution (MEXT, Japan, Tokyo) and the Guide for the Care and Use of Laboratory Animals of the Institute of Laboratory Animal Resources (ILAR; Washington, DC, USA). All animal procedures were approved by the Animal Care and Use Committee of the Kobe Institute of RIKEN (MA2008-03-11).

Reviewer #1 (Public review): https://doi.org/10.7554/eLife.99940.4.sa1
Reviewer #2 (Public review): https://doi.org/10.7554/eLife.99940.4.sa2
Author response https://doi.org/10.7554/eLife.99940.4.sa3

## Additional files

### Supplementary files

MDAR checklist

### Data availability

The 24-channel macaque coil is commercially available (Rogue Research, Montreal, Canada; manufactured by Takashima Seisakusho Co Ltd, Tokyo, Japan), and the data acquisition protocols are partially accessible from https://brainminds-beyond.riken.jp/hcp-nhp-protocol. The HCP-NHP

analysis pipelines are available through GitHub (*Brown et al., 2021*). The data presented in *Figures 4–6* have been deposited in the BALSA data repository (study ID: 1vjnV). Additional data, including measures of neuron and receptor densities as well as dendritic tree size and dendritic spines per layer 3 pyramidal cell are associated with the following study from BALSA (study ID: P2Nql).

The following dataset was generated:

| Author(s) | Year | Dataset title | Dataset URL | Database and Identifier |
|---|---|---|---|---|
| Autio JA, Kimura I, Ose T, Matsumoto Y, Ohno M, Urushibata Y, Ikeda T, Glasser MF, Van Essen DC, Hayashi T | 2025 | Mapping vascular network architecture in primate brain using ferumoxytol-weighted laminar MRI | https://balsa.wustl.edu/study/1vjnV | Brain Analysis Library of Spatial maps and Atlases, 1vjnV |

The following previously published dataset was used:

| Author(s) | Year | Dataset title | Dataset URL | Database and Identifier |
|---|---|---|---|---|
| Froudist-Walsh S, Xu T, Niu M, Rapan L, Zhao L, Margulies DS, Zilles K, Wang XJ, Palomero-Gallagher N | 2023 | Gradients of neurotransmitter receptor expression in the macaque cortex | https://balsa.wustl.edu/study/P2Nql | Brain Analysis Library of Spatial maps and Atlases, P2Nql |

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
