## [Editor Report · eLife Assessment]

This study presents **valuable** findings on the relative cerebral blood volume of non-human primates that move us closer to uncovering the functional and architectonic principles that govern the interplay between neuronal and vascular networks. The evidence of areal variations and of vessel counting and laminar analysis is **solid**. The lack of a direct comparison of their approach against better-established MRI-based methods for measuring hemodynamics and vascular structure somewhat weakens the evidence provided in the current paper version, but the current work is an significant step forward. The work will be of interest to NHP imaging scientists.

---

## [Referee Report · Reviewer #1 (Public review)]

Summary:

Audio et al. present an interesting study examining cerebral blood volume (CBV) across cortical areas and layers in non-human primates (NHPs) using high-resolution MRI. While with contrast agents are frequently employed to improve fMRI sensitivity in NHP research, its application for characterizing baseline CBV distribution is less common. This study quantifies large-vessel distribution as well as regional and laminar CBV variations, comparing them with other metrics.

Strengths:

(1) Noninvasive mapping of relative cerebral blood volume is novel for non-human primates.

(2) A key finding was the observation of variations in CBV across regions; primary sensory cortices had high CBV, whereas other higher areas had low CBV.

(3) The measured relative CBV values correlated with previously reported neuronal and receptor densities, potentially providing valuable physiological insights.

Weaknesses:

(1) A weakness of this manuscript is that the quantification of CBV with postprocessing approaches to remove susceptibility effects from pial and penetrating vessels is not fully validated, especially on a laminar scale.

(2) High-resolution MRI with a critical sampling frequency estimated from previous studies (Weber 2008, Zheng 1991) was performed to separate penetrating vessels. However, this approach depends on multiple factors, including spatial resolution, contrast agent dosage, and data processing methods. This raises concerns about the generalizability of these findings to other experimental setups or populations.

(3) Baseline R2* is sensitive to baseline R2, vascular volume, iron content, and susceptibility gradients. Additionally, it is sensitive to imaging parameters; higher spatial resolution tends to result in lower R2* values (closer to the R2 value). Although baseline R2* correlates with several physiological parameters, drawing direct physiological inferences from it remains challenging.

(4) CBV-weighted deltaR2*, which depends on both CBV and contrast agent dose, correlates with various metrics (cytoarchitectural parcellation, myelin/receptor density, cortical thickness, CO, cell-type specificity, etc.). While such correlations may be useful for exploratory analyses, all comparisons depend on measurement accuracy. A fundamental question remains whether CBV-weighted ΔR2* can provide reliable and biologically meaningful insights into these metrics, particularly in diseased or abnormal brain states.

---

## [Referee Report · Reviewer #2 (Public review)]

Summary:

This manuscript presents a new approach for non-invasive, MRI-based, measurements of cerebral blood volume (CBV). Here, the authors use ferumoxytol, a high-contrast agent and apply specific sequences to infer CBV. The authors then move to statistically compare measured regional CBV with known distribution of different types of neurons, markers of metabolic load and others. While the presented methodology captures and estimated 30% of the vasculature, the authors corroborated previous findings regarding lack of vascular compartmentalization around functional neuronal units in the primary visual cortex.

Strengths:

Non-invasive methodology geared to map vascular properties in vivo.

Implementation of a highly sensitive approach for measuring blood volume.

Ability to map vascular structural and functional vascular metrics to other types of published data.

Weaknesses:

The key issue here is the underlying assumption about the appropriate spatial sampling frequency needed to captures the architecture of the brain vasculature. Namely, ~7 penetrating vessels / mm2 as derived from Weber et al 2008 (Cer Cor). The cited work, begins by characterizing the spacing of penetrating arteries and ascending veins using vascular cast of 7 monkeys (*Macaca mulatta*, same as in the current paper). The ~7 penetrating vessels / mm2 is computed by dividing the total number of identified vessels by the area imaged. The problem here is that all measurements were made in a "non-volumetric" manner and only in V1. Extrapolating from here to other brain areas is therefore not possible without further exploration with independent methodologies.

Please note that these are comments on the revised version.

---

## [Author Response]

The following is the authors’ response to the previous reviews

**Reviewer #1 (Public review):**
Summary:Audio et al. measured cerebral blood volume (CBV) across cortical areas and layers using high-resolution MRI with contrast agents in non-human primates. While the non-invasive CBV MRI methodology is often used to enhance fMRI sensitivity in NHPs, its application for baseline CBV measurement is rare due to the complexities of susceptibility contrast mechanisms. The authors determined the number of large vessels and the areal and laminar variations of CBV in NHP, and compared those with various other metrics.Strengths:Noninvasive mapping of relative cerebral blood volume is novel for non-human primates. A key finding was the observation of variations in CBV across regions; primary sensory cortices had high CBV, whereas other higher areas had low CBV. The measured CBV values correlated with previously reported neuronal and receptor densities.

We appreciate your recognition of the novelty of our non-invasive relative cerebral blood volume (CBV) mapping in non-human primates, as well as the observed areal variations and their correlations with neuronal and receptor densities. However, we are concerned that key contributions of our work—such as cortical layer-specific vasculature mapping and benchmarking surface vessel density estimations against anatomical ground truth—are being framed as limitations rather than significant advances in the field pushing the boundaries of current neuroimaging capabilities and providing a valuable foundation for future research. Additionally, we would like to clarify that dynamic susceptibility contrast (DSC) MRI using gadolinium is the gold standard for CBV measurement in clinical settings and the argument that “baseline CBV measurements are rare due to the complexities of susceptibility contrast” is simply not true. The limited use of ferumoxytol for CBV imaging is primarily due to previous FDA regulatory restrictions, rather than inherent methodological shortcomings.

Changes in text:

Compared to clinically used gadolinium-based agents, ferumoxytol's substantially longer half-life and stronger R_2_* effect allows for higher-resolution and more sensitive vascular volume measurements (Buch et al., 2022), albeit these methodologies are hampered by confounding factors such as vessel orientation relative to the magnetic field (B_0_) direction (Ogawa et al., 1993).

Weaknesses:A weakness of this manuscript is that the quantification of CBV with postprocessing approaches to remove susceptibility effects from pial and penetrating vessels is not fully validated, especially on a laminar scale. Further specific comments follow.(1) Baseline CBV indices were determined using contrast agent-enhanced MRI (deltaR_2_*). Although this approach is suitable for areal comparisons, its application at a laminar scale poses challenges due to significant contributions from large vessels including pial vessels. The primary concern is whether large-vessel contributions can be removed from the measured deltaR_2_* through processing techniques.

Eliminating the contribution of large vessels completely is unlikely, and we agree with the reviewer that ΔR_2_* results likely reflect a weighted combination of signals from both large vessels and capillaries. However, the distribution of ΔR_2_* more closely aligns with capillary density in areas V1–V5 than with large vessel distributions (Weber et al., 2008), suggesting that our ΔR_2_* results are more weighted toward capillaries. Moreover, we demonstrated that the pial vessel induced signal-intensity drop-outs are clearly limited to the superficial layers and exhibit smaller spatial extent than generally thought (Supp. Figs. 2 and 4).

(2) High-resolution MRI with a critical sampling frequency estimated from previous studies (Weber 2008, Zheng 1991) was performed to separate penetrating vessels. However, this approach is still insufficient to accurately identify the number of vessels due to the blooming effects of susceptibility and insufficient spatial resolution. The reported number of penetrating vessels is only applicable to the experimental and processing conditions used in this study, which cannot be generalized.

Our intention was not to suggest that our measurements provide a general estimate of vessel density across the macaque cerebral cortex. At 0.23 mm isotropic resolution, we successfully delineated approximately 30% of the penetrating vessels in V1. Our primary objective was to demonstrate a proof-of-concept quantifiable measurement rather than to establish a generalized vessel density metric for all brain regions. We have consistently emphasized this throughout the manuscript, but if there is a specific point of misunderstanding, we would be happy to consider revisions for clarity.

(3) Baseline R_2_* is sensitive to baseline R_2_, vascular volume, iron content, and susceptibility gradients. Additionally, it is sensitive to imaging parameters; higher spatial resolution tends to result in lower R_2_* values (closer to the R_2_ value). Thus, it is difficult to correlate baseline R_2_* with physiological parameters.

The observed correlation between R_2_* and neuron density is likely indirect, as R_2_* is strongly influenced by iron, myelin, and deoxyhemoglobin densities. However, the robust correlation between R_2_* and neuron density, peaking in the superficial layers (R = 0.86, p < 10^-10^), is striking and difficult to ignore (revised Supp. Fig. 6D-E). Upon revision, we identified an error in Supp. Fig. 6D-E, where the previous version used single-subject R_2_* and ΔR_2_* maps instead of the group-averaged maps. The revised correlations are slightly stronger than in the earlier version.

Given that the correlation between neuron density and R_2_* is strongest in the superficial layers, we suggest this relationship reflects an underlying association with tissue cytochrome oxidase (CO) activity and cumulative effect of deoxygenated venous blood drainage toward the pial network. The superficial cortical layers are also less influenced by myelin and iron densities, which are more concentrated in the deeper cortical layers. Additional factors may contribute to this relationship, including the iron dependence of mitochondrial CO activity, as iron is an essential component of CO’s heme groups. Moreover, myelin maintenance depends on iron, which is predominantly stored in oligodendrocytes. The presence of myelinated thin axons and a higher axonal surface density may, in turn, be a prerequisite for high neuron density.

In this context, it is also valuable to note the absolute range of superficial R_2_* values (≈ 6 s^-1^; Supp. Fig. 6D). This variation in cortical surface R_2_* is about 12-30 times larger compared to the signal changes observed during task-based fMRI (6 vs. 0.2-0.5 s^-1^). This relation seems reasonable because regional increases in absolute blood flow associated with imaging signals, as measured by PET, typically do not exceed 5%–10% of the brain's resting blood flow (Raichle and Mintum 2016; Brain work and brain imaging). The venous oxygenation level is typically 60%, with task-induced activation increasing it by only a few percent. We suggest that this is ~40% oxygen extraction is reflected in the superficial R_2_*. Finally, the large intercept (≈ 14.5 1/s; Supp. Fig. 6D), which is not equivalent to the water R_2_* (≈ 1 1/s), suggests that R_2_* is influenced by substantial non-neuron density factors, such as receptor, myelin, iron, susceptibility gradients and spatial resolution.

The R_2_* values are well known to be influenced by intra-voxel phase coherence and thus spatial resolution. However, our view is that the proposed methodology of acquiring cortical-layer thickness adjusted high-resolution (spin-echo) R_2_ maps poses more methodological limitations and is less practical. Notwithstanding, to further corroborate the relationship between R_2_* and neuron density, we investigated whether a similar correlation exists in non-quantitative T2w SPACE-FLAIR images (0.32 mm isotropic) signal-intensity and neuron density. Using B_1_ bias-field and B_0_ orientation bias corrected T2w SPACE-FLAIR images (N=7), we parcellated the equivolumetric surface maps using Vanderbilt sections. Our findings showed that signal intensity—where regions with high signal intensity correspond to low R_2_ values, and areas with low signal intensity correspond to high R_2_ values—was positively correlated with neuron density, particularly in the superficial layers (R = 0.77, p = 10^-11^; Author response image 1).This analysis confirmed the correlation with neuron density and R_2_ peaks at superficial layers. However, this correlation was slightly weaker compared to quantitative R_2_* (Supp. Fig. 6D), suggesting the variable flip-angle spin-echo train refocused signal-phase coherence loss from large draining vessels or that non-quantitative T2w-FLAIR images may be confounded by other factors such as B_1_ transmission field biases (Glasser et al., 2022). Notwithstanding, this non-quantitative fast spin-echo with variable flip-angles approach, which is in principle less dependent on image resolution and closer to R_2,intrinsic_ than R_2_*, yields similar findings in comparison to quantitative gradient-echo.

**Author response image 1. sa3fig1:** (A) T2w-FLAIR SPACE normalized signal-intensity plotted vs neuron density. Note that low signal-intensity corresponds to high R_2_ and high neuron density, consistent with findings using ME-GRE. (B) Correlation between T2w-FLAIR SPACE and neuron density across equivolumetric layers. Notably, a similar relationship with neuron density was observed using a variable spin-echo pulse sequence as with quantitative gradient-echo-based imaging.

Changes in text:

Results:

“Because the Julich cortical area atlas covers only a section of the cerebral cortex, and the neuron density estimates are interpolated maps, we extended our analysis using the original Collins sample borders encompassing the entire cerebral cortex (Supp. Fig. 6A-C). This analysis reaffirmed the positive correlation with ΔR_2_* (peak at EL2, R = 0.80, *p* < 10^-11^) and baseline R_2_* (peak at EL2a, R = 0.86, *p* < 10^-13^), yielding linear coefficients of ΔR_2_* = 102 × 10^3^ neurons/s and R_2_* = 41 × 10^3^ neurons/s (Supp. Fig. 6D-G). This suggests that the sensitivity of quantitative layer R_2_* MRI in detecting neuronal loss is relatively weak, and the introduction of the Ferumoxytol contrast agent has the potential to enhance this sensitivity by a factor of 2.5.”

A new paragraph was added into discussion section 4.3 corroborating the relation between R_2_* and neuron density:

“Another key finding of this study was the strong correlation between baseline R_2_* and neuron density (Supp. Fig. 6D, E). While R_2_* is well known to be influenced by iron, myelin, and deoxyhemoglobin densities, this correlation peaks in the superficial layers (Supp. Fig. 6E), suggesting a link to CO activity and the accumulation of deoxygenated venous blood draining from all cortical layers toward the pial network. Notably, the absolute range of superficial R_2_* values (max - min ≈ 6 s^-1^; Supp. Fig. 6D) is approximately 12-30 times larger than the ΔR_2_* observed during task-based BOLD fMRI at 3T (0.2-0.5 1/s) (Yablonskiy and Haacke 1994). Since venous oxygenation is around 60% and task-induced changes in blood flow account for only 5%–10% of the brain's resting blood flow (Raichle & Mintun, 2006), these results suggest that superficial R_2_* (Fig. 1D) may serve as a more accurate proxy for total deoxyhemoglobin content (and thus total oxygen consumption), which scales with the neuron density of the underlying cortical gray matter. Importantly, superficial layers may also provide a more specific measure of deoxyhemoglobin, as they are less influenced by myelin and iron, which are more concentrated in deeper cortical layers. Additionally, smaller but direct contributors, such as mitochondrial CO density—an iron-dependent factor—may also play a role in this relationship.”

References:

Raichle, M.E., Mintun, M.A., 2006. BRAIN WORK AND BRAIN IMAGING. Annu. Rev. Neurosci. 29, 449–476. https://doi.org/10.1146/annurev.neuro.29.051605.112819

(4) CBV-weighted deltaR_2_* is correlated with various other metrics (cytoarchitectural parcellation, myelin/receptor density, cortical thickness, CO, cell-type specificity, etc.). While testing the correlation between deltaR_2_* and these other metrics may be acceptable as an exploratory analysis, it is challenging for readers to discern a causal relationship between them. A critical question is whether CBV-weighted deltaR_2_* can provide insights into other metrics in diseased or abnormal brain states.

We acknowledge that having multivariate analysis using dense histological maps would be valuable to establish causality among these several metrics:

“To comprehensively understand the factors contributing to the vascular organization of the brain, experimental disentanglement through multivariate analysis of laminar cell types and receptor densities is needed (Hayashi et al., 2021, Froudist-Walsh et al., 2023). Moreover, employing more advanced statistical modeling, including considerations for synapse-neuron interactions, may be important for refined evaluations.”

We think the primary contributors to the brain's energy budget are neurons and receptors, as shown in several references and stated in the manuscript. To investigate relationship between neuron density and CBV, we estimated the energy budget allocated to neurons and extrapolated the remaining CBV to other contributing factors:

Changes in text:

“However, this is a simplified estimation, and a more comprehensive assessment would need to account for an aggregate of biophysical factors such as neuron types, neuron membrane surface area, firing rates, dendritic and synaptic densities (Fig. 6F-G), neurotransmitter recycling, and other cell types (Kageyama 1982; Elston and Rose 1997; Perge et al., 2009; Harris et al., 2012). Indeed, the majority of the mitochondria reside in the dendrites and synaptic transmission is widely acknowledged to drive the majority of the energy consumption and blood flow (Wong-Riley, 1989; Attwell et al., 2001).

Extrapolating cortical ΔR_2_* to zero neuron density results in a large intercept (~35 1/s), corresponding to 60% of the maximum cortical CBV (57 1/s; Supp. Fig. 6F). This supports the view that the majority of energy consumption occurs in the neuropil—comprising dendrites, synapses, and axons—which accounts for ~80–90% of cortical gray matter volume, whereas neuronal somata constitute only ~10–20% (Wong-Riley, 1989). Although neuronal cell bodies exhibit higher CO activity per unit volume due to their dense mitochondrial content, these results suggest their overall contribution to the total CBV per mm^3^ tissue remains lower than that of the neuropil, given the latter's substantially larger volume fraction in cortical tissue.

Contrary to our initial expectations, we observed a relatively smaller CBV in regions and layers with high receptor density (Fig. 6B, D, F). This relationship extends to other factors, such as number of spines (putative excitatory inputs) and dendrite tree size across the entire cerebral cortex (Supp. Fig. 7) (Froudist-Walsh et al., 2023, Elston 2007). These results align with the work of Weber and colleagues, who reported a similar negative correlation between vascular length density and synaptic density, as well as a positive correlation with neuron density in macaque V1 across cortical layers (Weber et al., 2008).”

Variations in neurons and receptors are reflected in cytoarchitecture, myelin (axon density likely scales with neuron density and myelin inhibits synaptic connections), and cell-type composition. For example, fast-spiking parvalbumin interneurons, which target the soma or axon hillock, are well-suited for regulating activity in regions with high neuron density, whereas bursting calretinin interneurons, which target distal dendrites, are more adapted to areas with high synaptic density. These factors in turn, gradually change along the cortical hierarchy level (higher levels have thinner cortical layer IV, more complex dendrite trees and more numerous inter-areal connectivity patterns). In our view, these factors are tightly interlinked and explain the strong correlations and metabolic demands observed across different metrics.

We also agree that cortical layer imaging of vasculature in diseased or abnormal brain states is an intriguing direction for future research; however, it falls beyond the scope of the present study.

**Reviewer #2 (Public review):**
Summary:This manuscript presents a new approach for non-invasive, MRI-based, measurements of cerebral blood volume (CBV). Here, the authors use ferumoxytol, a high-contrast agent and apply specific sequences to infer CBV. The authors then move to statistically compare measured regional CBV with known distribution of different types of neurons, markers of metabolic load and others. While the presented methodology captures and estimated 30% of the vasculature, the authors corroborated previous findings regarding lack of vascular compartmentalization around functional neuronal units in the primary visual cortex.Strengths:Non invasive methodology geared to map vascular properties in vivo.Implementation of a highly sensitive approach for measuring blood volume.Ability to map vascular structural and functional vascular metrics to other types of published data.Weaknesses:The key issue here is the underlying assumption about the appropriate spatial sampling frequency needed to captures the architecture of the brain vasculature. Namely, ~7 penetrating vessels / mm2 as derived from Weber et al 2008 (Cer Cor). The cited work, begins by characterizing the spacing of penetrating arteries and ascending veins using vascular cast of 7 monkeys (*Macaca mulatta*, same as in the current paper). The ~7 penetrating vessels / mm2 is computed by dividing the total number of identified vessels by the area imaged. The problem here is that all measurements were made in a "non-volumetric" manner and only in V1. Extrapolating from here to the entire brain seems like an over-assumption, particularly given the region-dependent heterogeneity that the current paper reports.

We appreciate the reviewer’s concerns regarding spatial sampling frequency and its implications for characterizing brain vasculature, which we investigated in this study. To clarify, our analysis of surface vessel density was explicitly restricted to V1 precisely due to the limitations of our experimental precision. While we reported the total number of vessels identified in the cortex, we intentionally chose not to present density values across regions in this manuscript. Although these calculations are feasible, we focused on the data directly analyzed and avoided extrapolating density values beyond the scope of our findings. Thus, we are uncertain about the suggestion that we extrapolated vessel density values across the entire brain, as we have taken care to limit our conclusions of our vessel density precision to V1.

Regarding methodology, we conducted two independent analyses of vessel density specifically in V1. The first involved volumetric analysis using the Frangi filter, while the second used surface-based analysis of local signal-intensity gradients (as illustrated in Fig. 2E and Supp. Figs. 3 and 4), albeit the final surface density analysis is performed using the ultra-high resolution equivolumetric layers. Notably, these two approaches produced consistent and comparable vessel density estimates, supporting the reliability of our findings within the scope of V1 (we found 30% of the vessels relative to the ground-truth).

Comments on revisions:I appreciate the effort made to improve the manuscript. That said, the direct validation of the underlying assumption about spatial resolution sampling remains unaddressed in the final version of this manuscript. With the only intention to further strengthen the methodology presented here, I would encourage again the authors to seek a direct validation of this assumption for other brain areas.In their reply, the authors stated "... line scanning or single-plane sequences, at least on first impression, seem inadequate for whole-brain coverage and cortical surface mapping. ". This seems to emanate for a misunderstanding as the method could be used to validate the mapping, not to map per-se.

We apologize for any misunderstanding in our previous response and appreciate your clarification. We now understand that you were suggesting the use of line-scanning or single-plane sequences as a method to validate, rather than map, our spatial sampling assumptions.

We agree that single-plane sequences at very high in-plane resolution (e.g., 50 × 50 × 1000 µm) have great potential to detect penetrating vessels and even vessel branching patterns. These techniques could indeed provide valuable insights into region-specific vessel density variations which could then be used to validate whole brain 3D acquisitions. However, as noted above, we have refrained from reporting vessel densities outside V1 precisely due to sampling limitations (we only found 30% of the penetrating vessels in V1, or only 2 mm^2^/30mm^2^ ≈ 7% of branching vessel ground-truth, see discussion).

We acknowledge the merit of incorporating such methods to validate regional vessel densities and agree that this would be an important avenue for future research. Thank you for suggesting this point, we have briefly mentioned the advantage of single-plane EPI at discussion.

Changes in text:

“4.1 Methodological considerations - vessel density informed MRI

…anatomical studies accounting for branching patterns have reported much higher vessel densities up to 30 vessels/mm^2^ (Keller et al., 2011; Adams et al., 2015). Further investigations are warranted, taking into account critical sampling frequencies associated with vessel branching patterns (Duverney 1981), and achieving higher SNR through ultra-high B_0_ MRI (Bolan et al., 2006; Harel et al., 2010; Kim et al., 2013) and utilize high-resolution single-plane sequences and prospective motion correction schemes to accurately characterize regional vessel densities. Such advancements hold promise for improving vessel quantification, classifications for veins and arteries and constructing detailed cortical surface maps of the vascular networks which may have diagnostic and neurosurgical utilities (Fig. 2A, B) (Iadecola, 2013; Qi and Roper, 2021; Sweeney et al., 2018).”

During the revision we found a typo and corrected it in Supp. Fig. 8: Dosal -> Dorsal.